# Fast Non-Episodic Finite-Horizon RL with K-Step Lookahead Thresholding

**Jiamin Xu** [1]   **Kyra Gan** [1]

## Abstract

Online reinforcement learning in non-episodic, finite-horizon MDPs remains underexplored and is challenged by the need to estimate returns to a fixed terminal time. Existing infinite-horizon methods, which often rely on discounted contraction, do not naturally account for this fixed-horizon structure. We introduce a modified Q-function: rather than targeting the full-horizon, we learn a K-step lookahead Q-function that truncates planning to the next K steps. To further improve sample efficiency, we introduce a thresholding mechanism: actions are selected only when their estimated K-step lookahead value exceeds a time-varying threshold. We provide an efficient tabular learning algorithm for this novel objective, proving it achieves fast finite-sample convergence: it achieves minimax optimal constant regret for $K = 1$ and $\mathcal{O}(\max((K-1), C_{K-1})\sqrt{SAT \log(T)})$ regret for any $K \geq 2$. We numerically evaluate the performance of our algorithm under the objective of maximizing reward. Our implementation adaptively increases K over time, balancing lookahead depth against estimation variance. Empirical results demonstrate superior cumulative rewards over state-of-the-art tabular RL methods across synthetic MDPs and RL environments: JumpRiverswim, FrozenLake, and AnyTrading. Code is provided on github.

## 1. Introduction

In many real-world sequential decision problems—from medical treatment regimens to financial trading sessions (Liu et al., 2020; Trella et al., 2025; Ghosh et al., 2024; Hambly et al., 2023; Liu et al., 2021)—an agent must learn to perform well within a single, finite, and non-repeating trajectory. Formally, these are *non-episodic, finite-horizon*

Markov Decision Processes (MDPs), where the goal is to maximize cumulative reward up to a known terminal time without the benefit of environment resets.

Reinforcement learning (RL) theory, however, has largely developed around settings that differ fundamentally from this challenging regime. RL (Burnetas & Katehakis, 1997; Sutton et al., 1998) has been studied under two predominant settings: (1) infinite-horizon RL, where algorithm performance is evaluated under either the *average reward* (Boone & Zhang, 2024; Agrawal & Jia, 2017; Filippi et al., 2010; Fruit et al., 2020; Talebi & Maillard, 2018; Fruit et al., 2018; Bartlett & Tewari, 2012; Zhang & Xie, 2023; Wei et al., 2020; Agrawal & Agrawal, 2025; Abbasi-Yadkori et al., 2019) or *discounted cumulative reward* (Haarnoja et al., 2018; Mnih et al., 2015; Schulman et al., 2015; 2017) optimality criterion; and (2) episodic finite-horizon RL, where the environment resets to a known initial state after a fixed number of steps (Jin et al., 2018; Zhang et al., 2020; Xiong et al., 2022; Azar et al., 2017; Efroni et al., 2019). These algorithmic tools are generally categorized as *model-based*, which learn explicit transition dynamics to plan, or *model-free*, which learn value functions or policies directly.

Consequently, applying standard RL algorithms to *non-episodic finite-horizon* settings leads to poor finite-sample performance. Model-based methods (Boone & Zhang, 2024; Azar et al., 2017) are exponentially sample-inefficient in our setting as they must estimate the complete dynamics over the full horizon. Standard model-free, value-based methods (Zhang et al., 2020) face a fundamental barrier: they learn a Q-function that estimates returns until the terminal step. Without repeated episodes, accurately learning this full-horizon target from a single trajectory is inherently high-variance. Similarly, infinite horizon methods (Wei et al., 2020) require the operating horizon to exceed the MDP's diameter or mixing time to guarantee convergence. Within a fixed horizon, convergence to the optimal policy can not be assured. While deep learning extensions (Janner et al., 2019; Kaiser et al., 2019; Mnih et al., 2015; Schulman et al., 2017) enable scaling, their theoretical foundations—and thus their finite-sample behavior—rest on these tabular principles, making the tabular setting the necessary starting point for understanding and improving performance.

Our key insight is to address this estimation barrier by de-

---

[1]ORIE, Cornell Tech, New York, USA. Correspondence to: Kyra Gan <kyragan@cornell.edu>.

*Proceedings of the $43^{rd}$ International Conference on Machine Learning*, Seoul, South Korea. PMLR 306, 2026. Copyright 2026 by the author(s).

liberately limiting the planning depth. Instead of targeting the full-horizon Q-function, we propose learning a K-step lookahead Q-function paired with an adaptive thresholding rule. This reduces the complexity of the value target while maintaining alignment with the long-term objective. Our contributions are three-fold:

- We introduce the K-step lookahead thresholding policy, a new policy class designed for sample-efficient learning in non-episodic finite-horizon MDPs (Sec. 3). When $K$ exceeds the total horizon $T$, the policy is optimal, provided with a sufficiently high threshold. Although when $K < T$, the policy may exhibit an optimality gap linear in $T$, we prove that it is optimal for a two-state MDP under a stochastic dominance assumption (Assum. 3.2).

- We develop LGKT (*LCB-Guided K-Step Thresholding*, Algorithm 2, Sec. 4), a novel learning algorithm for non-episodic, finite horizon RL. We prove that when $K = 1$, LGKT achieves minimax optimal constant regret (Theorem 4.2), and for $K \geq 2$, its regret is $\mathcal{O}(\max((K-1), C_{K-1})\sqrt{SAT \log(T)})$ (Theorem 4.4) against the corresponding K-step benchmark where $C_{K-1}$ is an instance-dependent parameter (Sec. 4). This yields a sublinear *convergence rate* guarantee whenever $\max(K, C_{K-1})\sqrt{SA} = o(\sqrt{T})$, improving upon prior linear convergence results in our problem setting.

- We evaluate LGKT's cumulative reward performance against state-of-the-art tabular RL methods (Sec. 5) on a suite of *non-episodic environments*: 1) **1000 synthetic MDPs** with various state sizes, 2) **JumpRiverSwim** environments (Wei et al., 2020), and 3) **FrozenLake** (Brockman et al., 2016). LGKT consistently achieves the highest cumulative reward. Due to the lack of available non-episodic RL environments, we extend our evaluation to one *episodic environment* with continuous state spaces: **AnyTrading** (Haghpanah et al., 2023). LGKT again achieves the highest cumulative reward, demonstrating its effectiveness across diverse RL regimes.

### 1.1. Additional Related Work

While the idea of using an approximate policy as a sample-efficient oracle is closely related to work on restless bandits (RB) (Xu et al., 2026), our setting is fundamentally different. RB considers binary actions, state-dependent rewards, and a per-round budget constraint across agents, which reduces the problem to comparing the relative value of actions among agents. In contrast, we study general finite-horizon MDPs with arbitrary action spaces, state-and-action-dependent rewards, and no budget constraint. Consequently, our algorithm must evaluate the absolute value of each action, leading to a distinct algorithmic approach.

**Thresholding Bandit/MDP** Our paper relates to regret-minimizing thresholding bandit problems, under which min-

imax optimal (max over all possible bandit instances and min over all possible learning algorithms) constant regret is achievable (Tamatsukuri & Takahashi, 2019; Michel et al., 2022; Feng et al., 2025). Our problem setup differs from the traditional thresholding bandit framework by considering the problem in MDP and the reward structure. Hajiabol-hassan & Ortner (2023) study infinite-horizon thresholding MDPs, seeking policies with average rewards exceeding a predefined threshold. Our algorithm chooses actions whose K-step lookahead reward exceeds a threshold to improve finite-horizon performance.

**Lookahead in RL** Our work is distinct from two lines of research that also employ lookahead: (i) MDPs where the agent observes future rewards (Merlis, 2024; Pla et al., 2026; Merlis et al., 2024), and (ii) multi-step lookahead value/policy iteration based on a K-step Bellman operator in episodic (Efroni et al., 2020) and infinite-horizon (Protopapas & Barakat, 2024; Efroni et al., 2018; Bonet & Geffner, 2000) settings. In contrast, we use lookahead to denote a truncated Q-function for sample-efficient online learning in non-episodic, finite-horizon settings.

**Approximate Oracle MDP Solutions** With known MDP models, approximate solutions are used to mitigate the computational burden of planning in large state and action spaces. Bertsekas (2022) proposes K-step lookahead schemes that select actions by maximizing $\mathbb{E}[\sum_{t=0}^{K-1} R(s_t, a_t) + \tilde{V}(s_k) \mid a_0 = a]$, where $\tilde{V}$ is a user-specified lookahead function, and bounds the suboptimality gap of such policies in terms of the quality of $\tilde{V}$. De Farias & Van Roy (2003); Bertsekas & Ioffe (1996) used linear function approximation of the value functions and Munos (2003) proposed approximate policy iteration. Theoretical guarantees are provided in a discounted reward setting, relying on discounted contraction. While these solutions have inspired more efficient online learning algorithms (Abbasi-Yadkori et al., 2019), these algorithms still target the optimal policy, creating a fundamental learning bottleneck (Sec. 2.1). In contrast, we propose to learn a truncated Q-function, easing the learning objective. Our K-step lookahead thresholding policy shares a similar idea with K-step lookahead schemes (Bertsekas, 2022) with $\tilde{V} = 0$.

**Conflict of Interest Disclosure** We do not have any conflict of interest to disclose.

## 2. Problem Setup: Finite Horizon MDP

We consider finite-horizon MDPs with discrete states and actions. An MDP characterized by the tuple $(\mathcal{S}, \mathcal{A}, P_a, R)$, where $P_a(s, s') : \mathcal{S} \times \mathcal{S} \to [0, 1]$ denotes the transition probability from state $s$ to state $s'$ under action $a$. We will use $P_a(s) : \mathcal{S} \to \Delta(\mathcal{S})$ where $\Delta(\mathcal{S})$ denotes the probability simplex over all possible states to denote the distribution of

next state under state $s$ and action $a$. $R(s, a) : \mathcal{S} \times \mathcal{A} \to \mathbb{R}$ denotes the *expected reward* that a agent gets when in state $s$ and takes action $a$. We will use $R_{s,a}$ to denote $R(s, a)$ for simplicity. We will use $S := |\mathcal{S}|$ to denote the cardinality of $\mathcal{S}$ and $A := |\mathcal{A}|$ to denote the cardinality of $\mathcal{A}$.

At each time $t$, given the state $s_t$, the learner's decision $a_t$ follows a history-dependent random policy, $\pi_t$. Let $\mathcal{H}_t := \{(s_i, a_i, r_i)\}_{i=0}^{t-1} \cup \{s_t\}$ be the history observed up to time $t$. Similarly, denote $\mathcal{F}_t$ as the $\sigma$-algebra generated by $\mathcal{H}_t$. Then, $\pi_t : \mathcal{H}_t \to \Delta(\mathcal{A})$, where $\Delta(\mathcal{A})$ denotes the probability simplex over all possible actions. After pulling arm $a_t$, the learner receives a feedback $r_t$ and observes the next state $s_{t+1} \sim P_{a_t}(s_t, s_{t+1})$. Suppose that the reward $r_t$ is generated according to a distribution $\mathcal{P}_t$, we will assume stationarity of the *expected* reward received at time $t$, $r_t$, when conditioned on history $\mathcal{H}_t$ and the current action.

**Assumption 2.1.** $\mathbb{E}[r_t \mid a_t, \mathcal{H}_t] = R_{s_t, a_t}$.

We permit non-stationarity in the rewards, provided that Assumption 2.1 is satisfied, offering a relaxation of the standard stationarity assumption typically imposed in literature.

Given a fixed horizon of length $T$, the goal of the learner is to find the policy $\boldsymbol{\pi} := \{\pi_t\}_{t \in [T]}$ that maximizes the expected cumulative reward, where the expectation is further taken over the randomized policy ($a_t \sim \pi_t$), in addition to the states ($s_{t+1} \sim P_{a_t}(s_t, s_{t+1})$):

$$\max_{\boldsymbol{\pi}} \mathbb{E}_{\boldsymbol{\pi}} \left[ \sum_{t=0}^{T} R_{s_t, a_t} \mid s_0 \right]. \tag{1}$$

We will use $\boldsymbol{\pi}^*$ to denote the optimal policy corresponding to Problem (1).

We use $V_h^{\boldsymbol{\pi}}(s) : \mathcal{S} \to \mathbb{R}$ to denote the expected sum of remaining rewards received under policy $\boldsymbol{\pi}$ starting from state $s$ at time $h$ until the end of the horizon: $V_h^{\boldsymbol{\pi}}(s) := \mathbb{E}_{\boldsymbol{\pi}} \left[ \sum_{t=h}^{T} R_{s_t, a_t} \mid s_h = s \right]$. Similarly, we use $Q_h^{\boldsymbol{\pi}}(s, a) : \mathcal{S} \times \mathcal{A} \to \mathbb{R}$ to denote the expected sum of remaining rewards received under policy $\boldsymbol{\pi}$ starting from state $s$ at time $h$ until the end of the horizon: $Q_h^{\boldsymbol{\pi}}(s, a) := \mathbb{E}_{\boldsymbol{\pi}} \left[ \sum_{t=h}^{T} R_{s_t, a_t} \mid s_h = s, a_h = a \right]$. Let $V_h^* := \max_{\boldsymbol{\pi}} V_h^{\boldsymbol{\pi}}$ and $Q_h^* := \max_{\boldsymbol{\pi}} Q_h^{\boldsymbol{\pi}}$. We have the following Bellman optimality equations:

$$V_h^*(s) = \max_a Q_h^*(s, a), \forall s \in \mathcal{S}$$

$$Q_h^*(s, a) = R_{s,a} + \mathbb{E}_{s' \sim P_a(s, \cdot)}[V_{h+1}^*(s')], \forall s \in \mathcal{S}.$$

For simplicity, we assume that at each time, the action $a_{s,t}^*$ that maximizes $Q_{T-t}^*(s)$ is *unique* for all $t \in [T]$. The optimal policy $\boldsymbol{\pi}^*$ can be described as:

$$\pi_t^*(a_t | s) = \begin{cases} 1 & a_t = \arg\max_a Q_t^*(s, a) \\ 0 & \text{otherwise} \end{cases}. \tag{2}$$

## 2.1. Hardness of Learning Finite Horizon MDP Online

Both model-based (Azar et al., 2017) and model-free methods (Jin et al., 2018) depend on estimating $Q_h^*$. However, since the $Q$-function in a finite horizon is measuring the cumulative returns until the end of the horizon, it is sample-inefficient to learn in a finite horizon, non-episodic setting. The best known regret upper bound (Azar et al., 2017; Zhang et al., 2020) for episodic setting translates to a linear regret in a non-episodic setting, where the regret is defined as:

$$\mathcal{R}_Q^{\boldsymbol{\pi}^*}(\boldsymbol{\pi}) := \mathbb{E}_{\boldsymbol{\pi}^*} \left[ \sum_{t=0}^{T} R_{s_t, a_t} \mid s_0 \right] - \mathbb{E}_{\boldsymbol{\pi}} \left[ \sum_{t=0}^{T} R_{s_t, a_t} \mid s_0 \right].$$

Furthermore, the regret lower bound is also $\Omega(T)$ (Azar et al., 2017; Auer et al., 2008), meaning that all online algorithms suffer from a linear convergence rate to $\boldsymbol{\pi}^*$.

*Remark* 2.2 (Hardness of Applying Infinite-horizon Algorithms). While infinite-horizon average-reward maximization can serve as an approximation to Problem (1) for large horizons $T$, existing online algorithms designed for this setting rely on structural assumptions that guarantee convergence to a stationary distribution (Wei et al., 2020) or quick recovery to high-reward states (Boone & Zhang, 2024). In a short, finite horizon, these underlying assumptions are not satisfied, and thus the guaranteed properties do not hold. As we demonstrate empirically in Section 5, this mismatch leads to slow convergence for such algorithms in our setting.

## 3. K-step Lookahead Q-function and K-step Lookahead Thresholding

As mentioned above, learning $\boldsymbol{\pi}^*$ suffers from slow convergence because of the need to estimate $Q_h^*$. To address this problem, we will make two key modifications:

1. *K-step Lookahead Q-function*: We target a K-step lookahead Q-function $Q_{T-K}^*$. This means instead of estimating the return until the end of the horizon, we only focus on learning the return in K steps. By truncating the horizon, sample complexity can be improved. When $K = 1$, the problem becomes a contextual bandit where context follows an MDP, which is known to have a lower sample complexity (Simchi-Levi & Xu, 2022).

2. *Thresholding objective*: At each step, we identify actions that the K-step lookahead reward exceeds a threshold (vs. optimizing cumulative rewards), further lowering sample complexity (Feng et al., 2025) to achieve fast finite-horizon convergence.

To formalize this approach, we first define the core quantity used for K-step planning: the lookahead reward.

**Definition 3.1** (K-Step Lookahead Reward). The K-step lookahead reward $r_{s,a}^{\mathrm{K}}$ is the maximum total expected reward attainable from state $s$ and action $a$ over the next $K$

steps. Due to the time-homogeneity of the MDP, this equals the optimal Q-value at step $T - K + 1$ (with exactly $K$ steps remaining): $r_{s,a}^{\mathrm{K}} := Q_{T-K+1}^*(s, a)$.

We denote by $a_{s,K}^*$ the action that maximizes $r_{s,a}^{K}$: $a_{s,K}^* := \arg\max_a r_{s,a}^{\mathrm{K}}$ for $K \geq 1$.

A K-step lookahead greedy policy $\boldsymbol{\pi}^{K,\mathrm{greedy}}$ makes decisions by looking ahead at most $K$ steps. When the remaining horizon $h = T - t + 1$ is greater than $K$, it chooses the action maximizing the $K$-step reward $r_{s,a}^{\mathrm{K}}$ with respect to $a$. When $h \leq K$, it looks ahead to the terminal step by maximizing the $h$-step reward $r_{s,a}^{\mathrm{h}}$ with respect to $a$. This logic is captured succinctly by maximizing $r_{s,a}^{\min(\mathrm{h},\mathrm{K})}$ with respect to $a$. Formally, $\boldsymbol{\pi}^{K,\mathrm{greedy}}$ is defined as:

$$\pi_t^{\mathrm{K,greedy}}(a_t|s) = \begin{cases} 1 & a_t = \arg\max_a r_{s,a}^{\min(\mathrm{h,K})} \\ 0 & \text{otherwise} \end{cases}. \quad (3)$$

To further reduce the sample complexity, we target at K-step lookahead thresholding policy, denoted by $\boldsymbol{\pi}^{\mathrm{K},\gamma}$. At each time step $t$, this policy selects an action uniformly at random from those satisfying $r_{s_t,a}^{\min(h,\mathrm{K})} \geq \gamma_t$. If no such action exists, it instead selects $a_{s,\min(h,K)}^*$. We note that by setting the thresholds $\boldsymbol{\gamma} := \{\gamma_t\}_{t \in [T]}$ sufficiently high, this policy reduces to a greedy policy.

**Regret as Convergence**   Following prior work (Xu et al., 2026; Candelieri, 2023), we use regret to measure the convergence rate of online algorithms to the oracle policy that they are converging to. Since our algorithm converges to the K-step lookahead thresholding policy $\pi^{K,\gamma}$ and not to the optimal policy $\pi^*$, the regret is consequently defined as:

$$\mathcal{R}^{\boldsymbol{\pi}^{\mathrm{K},\gamma}}(\boldsymbol{\pi}) := \mathbb{E}_{\boldsymbol{\pi}^{\mathrm{K},\gamma}} \left[ \sum_{t=0}^{T} c(s_t, a_t) \mid s_0 \right]$$
$$- \mathbb{E}_{\boldsymbol{\pi}^{\mathrm{K},\gamma}} \left[ \sum_{t=0}^{T} c(s_t, a_t) \mid s_0 \right], \quad (4)$$

where cost function $c(s_t, a_t)$ is defined as:

$$c(s_t, a_t) := (\gamma_t - r_{s_t,a_t}^{\min(\mathrm{h,K})}) \mathbb{I}\left\{ r_{s_t,a_t}^{\min(\mathrm{h,K})} < \gamma_t \right\}.$$

This cost function penalizes the selection of any "bad" action whose K-step lookahead reward falls below the threshold. Thus, Eq. (4) measures how frequently a policy $\pi$ deviates from the K-step lookahead thresholding policy $\boldsymbol{\pi}^{\mathrm{K},\gamma}$, which in turn qualifies the rate of convergence toward $\boldsymbol{\pi}^{\mathrm{K},\gamma}$.

### 3.1. Optimality of K-Step Lookahead Greedy Policy

When $K$ exceeds the horizon $T$, i.e., $K \geq T$, the greedy policy $\boldsymbol{\pi}^{K,\mathrm{greedy}}$ coincides with the optimal policy $\boldsymbol{\pi}^*$ by Eq. (3), and is therefore optimal.

In the rest of the subsection, we first establish that $\boldsymbol{\pi}^{K,\mathrm{greedy}}$ is always optimal for a *two-state* MDP under the stochastic dominance assumption (Assm. 3.2) in Thm. 3.3. Next, we show that in the general state setting, $\boldsymbol{\pi}^{K,\mathrm{greedy}}$ incurs an optimality gap that is linear in $T$ in the worst case (Thm. 3.4).

**Assumption 3.2** (Stochastic Dominance Assumption). Consider an MDP with a binary state space $\mathcal{S} = \{0, 1\}$, where state 1 yields higher maximum rewards: $\max_a r_{1,a} \geq \max_a r_{0,a}$. Let $a_{s,1}^*$ denote the action maximizing the one-step lookahead reward from state $s$. We assume that for every state $s \in \mathcal{S}$ and every action $a \in \mathcal{A}$, $P_{a_{s,1}^*}(s, 1) \geq P_a(s, 1)$.

Intuitively, Assumption 3.2 guarantees that the action with the highest immediate reward also maximizes the probability of transitioning to the more rewarding state. This alignment ensures that short-term greedy decisions do not compromise long-term value. In healthcare decision-making, Assumption 3.2 is satisfied: higher treatment intensity (action) can both increase immediate reward (e.g., higher recovery probability) and shift the next-state distribution toward healthier states. Theorem 3.3 establishes the optimality of $\boldsymbol{\pi}^{K,\mathrm{greedy}}$ for any $K \geq 1$.

**Theorem 3.3.** *Under Assumption 3.2 and binary states, $\boldsymbol{\pi}^{K,greedy} = \boldsymbol{\pi}^*$ for any $K \geq 1$.*

The proof (Appendix B.1) relies on the use of inductions to show that $V_t^*$ is a monotone function and then use Assumption 3.2 to show the optimality.

For general states, we show that there always exists an MDP in which the $K$-step lookahead thresholding policy incurs a linear optimality gap for any fixed $K < T$.

**Theorem 3.4.** *For any $K < T, \mathcal{S}, \mathcal{A}$, there exists an MDP instance with a state $s \in \mathcal{S}$ such that $V_0^{\boldsymbol{\pi}^*}(s) - V_0^{\boldsymbol{\pi}^{\mathrm{K},\gamma}}(s) = \Theta(T)$.*

The proof of Thm. 3.4 (Appendix B.2) relies on constructing a bad instance where the beneficial long-term action carries a large immediate penalty. While taking this penalty is optimal over the full horizon, a K-step lookahead policy (with $K < T$) will always reject it due to the short-term cost, thereby remaining trapped in a low-value state. This results in a per-step loss that accumulates linearly with $T$.

To better characterize the instance-dependent constant of the linear suboptimality gap, we show that the suboptimality gap is dependent on the $\mathcal{L}_1$ distance of transition probabilities.

**Theorem 3.5.** *For any $K < T, \mathcal{S}, \mathcal{A}$,*

$$V_0^*(s) - V_0^{\boldsymbol{\pi}^{\mathrm{K},\gamma}}(s) \quad (5)$$
$$\leq \sum_{t=0}^{T-K} \max_{s'} \left| P_{\pi_t^{\mathrm{K},\gamma}}(s') - P_{a_{s',T-t+1}^*}(s') \right|_1 \max_{s'} V_{t+1}^*(s'),$$

*where $P_{\pi_t^{\mathrm{K},\gamma}}(s') := \sum_a \pi_t^{\mathrm{K},\gamma}(a|s') P_a(s')$.*

The proof of Theorem 3.5 (Appendix B.3) relies on backward induction on the difference of value function. Theorem 3.5 shows that the K-step lookahead thresholding policy can achieve near optimal when the transition matrix under different actions are close in $\mathcal{L}_1$ distance.

Before we introduce our online learning algorithm in Section 4, we note that a fundamental trade-off exists between an algorithm's convergence rate and the optimality of its target policy. While Theorems 3.4 and 3.5 shows that $\pi^{K,\gamma}$ can have linear suboptimality, Section 4 will demonstrate that an online algorithm can achieve sublinear regret against this oracle. This stands in contrast to the known linear regret lower bound when competing with the optimal policy (Section 2). We argue that in a finite horizon, a fast convergence rate is the key to achieving high cumulative reward—a claim that we empirically validate in Section 5.

## 4. Online K-Step Lookahead

We begin by presenting our learning algorithm for the 1-step lookahead thresholding policy (Alg. 1, Sec. 4.1). We establish that Algorithm 1 achieves a minimax optimal constant regret (Thm. 4.2). Next, we extend it to the general K-step case (Alg. 2, Sec. 4.2). We establish that Algorithm 2 achieves a regret bound of $\max(\mathcal{O}((K-1)\sqrt{SAT}), \mathcal{O}(C_{K-1}\sqrt{SAT\log(T)}))$ (Thm. 4.4). These results demonstrate a fast convergence rate for our method when $K$ is small in the non-episodic finite-horizon setting (Remark 4.6). In this section, we *do not require* Assumption 3.2 or binary state.

### 4.1. Online One-Step Lookahead

When $K = 1$, our problem reduces to a standard thresholding contextual bandit problem. Algorithm 1 selects actions for the one-step lookahead policy via a lower confidence bound (LCB) test against the threshold $\gamma$. At each time $t$:

1. It constructs a candidate set $\tilde{G}_t$ of actions whose LCB for the one-step reward is at least $\gamma_t$.
2. If $|\tilde{G}_t| \geq 1$, it plays the action from this set with the highest LCB.
3. Otherwise, it plays an action uniformly at random.

After observing the reward $r_t$ and next state $s_{t+1}$, the algorithm updates the LCB for the observed state-action pair $(s, a)$. Let $\hat{r}^1_{s,a}(t+1)$ be the empirical mean reward and $N^{(t+1)}_{s,a}$ be the visit count. The $\text{LCB}^{(t+1)}_{s,a}$ is defined by

$\hat{r}^1_{s,a} - \sqrt{\frac{g\left(N_{s,a}+2\right)}{N_{s,a}+2}}$, with $g(t) = 3\log(t)$.

A key distinction from prior work (e.g., Algorithm 1 of Michel et al. (2022)) is our use of the LCB—rather than the empirical mean—to define $\tilde{G}_t$. Our choice of $g(t) = 3\log(t)$ is smaller than the typical anytime-valid LCB of

---

**Algorithm 1 LG1T**: LCB-Guided 1-Step Thresholding

1: **Input:** initial state $s$, exploration function $g(t), \gamma$
2: Initialize: $s_0 = s$, $\forall s \in \mathcal{S}, a \in \mathcal{A} : N^{(0)}_{s,a} = 0$, $\hat{\phi}(s,a) = 0, \text{LCB}^{(0)}_{s,a} = -\infty$
3: **for** $t = 0, 1, \cdots, T$ **do**
4:   Retrieve $s_t$, and obtain the set of good actions $\tilde{G}_t = \left\{ a \colon \text{LCB}^{(t)}_{s_t,a} \geq \gamma_t \right\}$. Set $C = \tilde{G}_t$;
5:   Select $\max(0, 1-|\tilde{G}_t|)$ action uniformly from $\mathcal{A} \setminus \tilde{G}_t$, and append them to $C$
6:   Select an action from $C$ according to $\text{LCB}^{(t)}$, and denote it by $a$. Set $a_t = a$
7:   Observe $(r_t, s_{t+1})$
8:   $\hat{\phi}(s,a), N^{(t+1)}_{s,a}, \text{LCB}^{(t+1)}_{s,a}$ =Update-LCB-1 $(\hat{\phi}(s,a), N^{(t)}_{s,a}, r_t, s_t, a_t)$ (Algorithm A.4)
9: **end for**

---

$g(t) = \Theta(\log(t))$. This ensures that the LCBs of truly sub-threshold actions drop below $\gamma_t$ rapidly, enabling faster identification and elimination of suboptimal actions. This is crucial for obtaining our improved regret bound in Theorem 4.2. Before stating our theorem, we first introduce the following assumption on the threshold $\gamma$.

**Assumption 4.1** (Exists A Good Action). Assume $\forall t \leq T$, $\forall s \in \mathcal{S}$, there exists an action $a \in \mathcal{A}$ such that $r^K_{s,a} \geq \gamma_t$.

Assumption 4.1 is independent of the structure of the MDP, and can be satisfied by choosing a low $\gamma$.

We will let $\Delta^K_{s,a,t} = \gamma_t - r^K_{s,a}$ denote the gap between the threshold and the K-step lookahead reward, and define $\Delta_K = \min_{s,a,t} |\Delta^K_{s,a,t}|$. Further, we define $\Delta^K_+ := \min_{\Delta^K_{s,a,t} \leq 0} |\Delta^K_{s,a,t}|$ as the smallest gap among actions whose one-step reward is above the threshold. Thm. 4.2 establishes that Algorithm 1 has a regret of $\mathcal{O}(SA/\Delta^1_+)$:

**Theorem 4.2.** *Under Assumption 4.1 with $K = 1$, the regret $\mathcal{R}^{\pi^{1,\gamma}}$ of Algorithm 1 satisfies $\mathcal{R}^{\pi^{1,\gamma}} = \min\left\{ \mathcal{O}\left(SA/\Delta^1_+\right), \mathcal{O}\left(\sqrt{SAT}\right) \right\}$.*

Thm. 4.2 aligns with the minimax lower bound established in Feng et al. (2025) (Thm. 3). When $S = 1$, Thm. 4.2 yields a regret of $\mathcal{O}\left(A/\Delta^1_+\right)$, which depends only on the gap between the one-step lookahead reward above the threshold and the threshold. This improves the regret bound of $\mathcal{O}\left(\max_{\Delta^1_{a,t}>0} 1/\Delta^1_{a,t}\right)$ from Michel et al. (2022), which depends on the gap between rewards below the threshold and the threshold—typically smaller than $\Delta^K_+$. The proof structure of Thm. 4.2 (in Appendix D.2) is similar to that of a standard contextual bandit, but shows that our LCB design eliminates suboptimal actions in constant time.

## 4.2. Online K-Step Lookahead

We now extend our approach to the general K-step lookahead thresholding policy, which provides greater flexibility for long-term planning. Learning the K-step lookahead thresholding policy is hard because the K-step reward $r^{\mathrm{K}}_{s_t,a_t}$ is not directly observed. We decompose it into two terms: $r^1_{s_t,a_t}$ and $\mathbb{E}\left[r^{\mathrm{K}-1}_{s_{t+1},a^*_{t+1,\mathrm{K}-1}}|s_t,a_t\right]$. To estimate the K$-1$-step reward, the agent must, after taking action $a_t$ in state $s_t$, follow the optimal K-step continuation $a^*_{s_{t+1},\mathrm{K}-1}$ from the next state $s_{t+1}$. However, playing $a^*_{s_{t+1},\mathrm{K}-1}$ may itself yield low reward and increase regret. This creates a tension between minimizing regret and collecting the information needed for accurate long-horizon estimation.

To mitigate this, we use an $\epsilon$-greedy framework. At each time $t$, with probability $\epsilon_t$ that inversely proportional to the number of times the learner spent in the previous state playing previous action $N^{(t)}_{s_{t-1},a_{t-1}}$, Algorithm 2 runs Estimate-$r^{\mathrm{K}-1}$ (Algorithm A.3), which uses a *given* algorithm $\mathrm{ALG}_{\mathrm{K}-1}(\cdot \mid s_{t-1},a_{t-1})$ to choose actions for the next $K-1$ consecutive steps and collect samples $\sum_{h=t}^{t+K-1} r_h$ to estimate $r^{\mathrm{K}-1}_{s_t,a^*_{s_t,\mathrm{K}-1}}$. $\mathrm{ALG}_{\mathrm{K}-1}$ takes the state-action pair $(s_{t-1},a_{t-1})$ as input and performs its own internal updates using collected data. These updates are maintained independently from the main algorithm's estimators. The choices of $\mathrm{ALG}_{\mathrm{K}-1}$ are discussed in Appendix C.1. With probability $1-\epsilon_t$, Algorithm 2 uses the same procedure as described in Algorithm 1 which uses LCB to determine whether the K-step lookahead reward is above the threshold. After playing the action, lower confidence bound is updated (Algorithm A.5).

Theoretical regret depends on the error of samples that is collected by $\mathrm{ALG}_{\mathrm{K}-1}(\cdot|s_{t-1},a_{t-1})$ to estimate $K-1$-step lookahead reward $r^{\mathrm{K}-1}_{s_t,a^*_{s_t,\mathrm{K}-1}}$. Formally, we make the following assumptions to bound the error.

**Assumption 4.3** (Exists A Good Sampling Policy). Given any state-action pair $(s,a)$ and a positive constant $H$, run $\mathrm{ALG}_{\mathrm{K}-1}(\cdot|s,a)$ for $H$ times and collect $H$ trajectories $\left\{(s_0^h,a_0^h,s_1^h,a_1^h,\cdots,s_{K-2}^h,a_{K-2}^h)\right\}_{h\in[H]}$ where $s_0^h \sim P_a(s)$. Then with probability $1-\delta$: $\mathbb{E}_{s'\sim P_a(s)}\left[r^{\mathrm{K}-1}_{s',a^*_{s',\mathrm{K}-1}}\right] - \sum_{h=1}^{H}\sum_{k=0}^{K-2} R_{s_k^h,a_k^h} = \mathcal{O}\left(\sqrt{C_{K-1}H\log(SAH/\delta)}\right)$.

We emphasize that Assumption 4.3 can be satisfied by existing algorithms. Details are deferred to Appendix C.1. Next, we present the regret of Algorithm 2.

**Theorem 4.4.** *Under Assumptions 4.1, 4.3, let* $\eta = \Delta^{\mathrm{K}}, p = 1/2, g(t) = \sqrt{\frac{3\log(t)}{t}}$, *Algorithm 2 admits the following regret for any K:* $\mathcal{R}^{\pi^{\mathrm{K},\gamma}} = \max\left\{\mathcal{O}\left((K-1)\sqrt{SAT}\right), \mathcal{O}\left(C_{K-1}\sqrt{SAT\log(T)}\right)\right\}$.

When $K = 2$, UCB algorithm satisfies Assumption 4.3 with $C_{K-1} = A$ (Appendix C.1). Therefore, we immediately have the following corollary.

**Corollary 4.5.** *For 2-step lookahead thresholding, let* $\mathrm{ALG}_1$ *be the UCB algorithm,* $\eta = \Delta^2/\sqrt{A}, p = 1/2, g(t) = \sqrt{\frac{3\log(t)}{t}}$, *Algorithm 2 has the following regret:* $\mathcal{R}^{\pi^2,\gamma} = \mathcal{O}\left(A\sqrt{ST\log(T)}\right)$.

*Remark* 4.6 (Impact of K On Convergence Rate). Thm. 4.4 demonstrates that when the number of steps lookahead increases, the convergence will be slower. Further, when $\max(K,C_{K-1}\sqrt{SA}) = o(\sqrt{T})$, Thm. 4.4 enjoys a sublinear convergence rate to $\pi^{\mathrm{K},\gamma}$, improving over the linear convergence rate to the optimal policy (Azar et al., 2017). This dependence on $K$ shows there is a trade-off between maximizing the convergence rate and maximizing the cumulative reward. Empirically, we show that $K = 1,2$ can already outperforms SOTA tabular RL algorithms for a long horizon for all instances we test (Section 5).

The proof of Thm. 4.4 (Appendix D.3) follows the structure of Thm. 4.2, with two key modifications: 1) Bounding *regret from exploration*, 2) Ensuring *sufficient estimation samples*. A proof sketch is provided in Appendix D.3.

Selecting the threshold $\gamma$ involves a fundamental trade-off. A low threshold ensures Assumption 4.1 and fast convergence (Theorem 4.2), but potentially selects from more actions, lowering cumulative reward. A higher threshold yields slower convergence but can converge to a better policy. Thus, we must balance between maximizing cumulative reward and maximizing convergence rate. Theoretically, a low $\gamma$ may perform better early, while a high $\gamma$ can eventually achieve higher reward. Nevertheless, as shown in Section 5, moderate values of $\gamma$ achieve strong performance, with ablation studies (Appendix E.6) indicating that results are often robust to this choice across many environments.

Further, Theorems 4.2 and 4.4 hold for time and state varying threshold. Therefore, one can start from a low threshold and increase the threshold gradually to achieve both fast convergence and asymptotic optimality. We provide one heuristic of doing so in Appendix E.3.

Before we show the empirical results, we discuss the regime where fast convergence translates to higher cumulative reward.

*Remark* 4.7 (Favorable Regime of Fast Convergence). As Theorem 4.4 shows, LGKT achieves sublinear regret compared with K-step lookahead thresholding policy. Since the regret lower bound compared with the exact optimal policy is linear (Section 2.1), in order to theoretically characterize the favorable regime of fast convergence, we only need to compare the regret lower bound of tracing exact optimal policy with the suboptimality gap of K-step lookahead

---

**Algorithm 2 LGKT**: LCB-Guided K-Step Thresholding

---

1: **Input:** initial state $s$, exploration function $g(t)$, threshold $\gamma$, instance parameter $\eta$, exploration decay rate $p$, sampling algorithm that takes in initial state-action pair: $\text{ALG}_K(\cdot|\text{initial state,initial action})$

2: Initialize: $s_0 = s, \forall (s,a) \in \mathcal{S} \times \mathcal{A} : N_{s,a}^{(0)} = N_{s,a,K-1}^{(0)} = \hat{\phi}_{s,a}^1 = \hat{\phi}_{s,a}^{K-1} = 0, \text{LCB}_{s,a}^{(0)} = -\infty$

3: **while** $t \leq T$ **do**

4:     Retrieve $s_t$, and obtain the set of good actions $\tilde{G}_t = \left\{ a : \text{LCB}_{s_t,a}^{(t)} \geq \gamma_t \right\}$. Set $C = \tilde{G}_t$;

5:     Select $\max(0, 1 - |\tilde{G}_t|)$ actions uniformly from $\mathcal{A} \setminus \tilde{G}_t$, and append them to $C$

6:     Select an action from $C$ according to the order of $\text{LCB}^{(t)}$, and denote it by $a$.

7:     **if** With probability $\epsilon_t = \min \left\{ 1, \frac{1}{\left( N_{s_{t-1},a_{t-1}}^{(t)} + 1 \right)^p \min\{\eta, 1/2\}} \right\}$: **then**

8:         $N_{s,a,K-1}^{(t+K-1)}, N_{s,a}^{(t+K-1)}, \hat{\phi}^{K-1}, \hat{\phi}^1, \text{LCB}^{(t+K-1)} = \text{Estimate-}r^{K-1}(\text{ALG}_{K-1}, s_{t-1}, a_{t-1}, s_t, t)$ (Algorithm A.3)

9:         $t = t + K - 1$

10:    **else**

11:        K-Step thresholding: set $a_t = a$ and observe $(r_t, s_{t+1})$

12:        $\hat{\phi}_{s,a}^1, N_{s,a}^{(t+1)}, N_{s,a,K-1}^{(t+1)}, \text{LCB}_{s,a}^{(t+1)} = \text{Update-LCB-K}(\hat{\phi}_{s,a}^1, \hat{\phi}_{s,a}^{K-1}, N_{s,a}^{(t)}, N_{s,a,K-1}^{(t)}, r_t, s_t, a_t)$ (Algorithm A.5)

13:        $t = t + 1$

14:    **end if**

15: **end while**

---

thresholding policy as characterized in Theorem 3.5.

# 5. Experiments

We evaluate our proposed algorithms—**LG1T** (Algorithm 1), **LG2T** (Algorithm 2), and an adaptive variant **LG1-2T** (Algorithm A.6)—across several environments. LG1-2T gradually increases the lookahead steps during learning, aiming to balance fast initial convergence with long-term reward maximization. Because standard benchmarks for non-episodic finite-horizon settings are scarce, we test our methods in both non-episodic and episodic regimes. Our non-episodic experiments include: (1) 2000 **synthetic MDPs** with discrete states, (2) **JumpRiversSwim** (Wei et al., 2020) with three different sets of state spaces, (3) **FrozenLake** (Brockman et al., 2016) ($4 \times 4$ map). We additionally test on an episodic environment with continuous state space, (4) **AnyTrading** (Haghpanah et al., 2023). We describe the environment details in Appendix E.1. Across all experiments, we set the total horizon length to be $20,000$. We set the threshold for LG1T to be 0.3 and LG2T to be 0.9. The LG1-2T algorithm switches from LG1T to LG2T at a specified change time, and we set the change time to 100 for Experiments (1) and (2), 10,000 for Experiment (3), and 30 for Experiment (4). An ablation on the choice of threshold is included in Appendix E.6. We provide a heuristic for adaptively tuning the threshold in Appendix E.3 and a heuristic for adaptively choosing the change time for LG1-2T in Appendix E.4. Our methods consistently achieve the highest cumulative reward across these tests.

**Benchmarks** We compare against **six** state-of-the-art **tabular RL algorithms**. Three model-based average-reward algorithms: 1) UCRL2 (Auer et al., 2008), 2) KLUCRL (Filippi et al., 2010), 3) PMEVI-KLUCRL (Boone & Zhang, 2024); three model-free average-reward algorithms: 4) MDP-OOMD (Wei et al., 2020), 5) Optimistic Q Learning (Wei et al., 2020) and episodic RL: 6) Q Learning (Jin et al., 2018). Hyperparameters for all benchmarks are set as reported in their original papers. We also evaluate **two oracle policies**: 1) 1-step lookahead greedy $\pi^{1,\text{greedy}}$ and 2) 2-step lookahead greedy $\pi^{2,\text{greedy}}$, and compute the optimality ratio of each policy relative to the full oracle policy. Descriptions of the benchmark algorithms along with implementation details are provided in Appendix E.2.

**Modifications** In all experiments, we modify LG1T and LG2T by replacing the uniform random selection (Line 5) with a UCB-based choice: actions are chosen in decreasing order of the index $\hat{r}_{s,a}(t) + \frac{3.4}{N_{s,a}^{(t)}} \sqrt{\frac{\log\log\left(N_{s,a}^{(t)}\right) + \log(10T)}{N_{s,a}^{(t)}}}$. This index uses an anytime-valid confidence bound (Howard et al., 2021) scaled by $1/N_{s,a}^{(t)}$, a heuristic scaling that discourages excessive exploration on poorly performing actions and accelerates convergence (Jin et al., 2018). We show in Appendix E.5 that though this modification can further improve the cumulative reward, the unmodified version still outperforms all benchmarks with a low threshold. This shows that the empirical gain comes from the thresholding idea and the use of LCB to test whether the reward is above threshold. Using UCB instead of uniform when no actions' LCB is above threshold mainly improves robustness and performance when the threshold is high. We further show in Appendix F that the modified Algorithm 1 achieves the same convergence rate up to log factor.

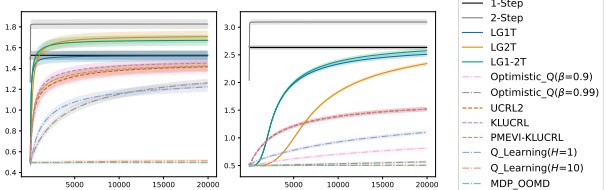

*Figure 1.* Running average reward over 1000 distinct MDP instances. Left: $S = 10$, $A = 5$, Right: $S = 100$, $A = 25$. Right excludes PMEVI-KLUCRL due to its prohibitive runtime and on Right, KLUCRL overlaps with UCRL2.

## 5.1. Discrete State Space

**Synthetic MDP** Figure 1 presents the results on 1,000 randomly generated MDPs with 10 states and 5 actions, and 1,000 with 100 states and 25 actions. Our algorithms (LG1T, LG2T, and LG1-2T) converge rapidly to their respective oracle policies, achieving the highest final performance. For the 10-state case, the average competitive ratios across 1000 instances are $V_0^{\pi^{1,\text{greedy}}}/V_0^* = 0.75$ and $V_0^{\pi^{2,\text{greedy}}}/V_0^* = 0.96$. The strong empirical performance of our algorithms, even when learning suboptimal oracles, demonstrates that existing RL methods suffer from slow convergence, while our approach quickly converges to high-quality policies.

In the 100-state case, the average competitive ratios are $0.8$ for $\pi^{1,\text{greedy}}$ and $0.94$ for $\pi^{2,\text{greedy}}$. PMEVI-KLUCRL is excluded due to its prohibitive runtime ($>3$ h/instance); its performance is expected to be similar to KLUCRL (Boone & Zhang, 2024). While model-based average-reward algorithms initially achieve higher reward, both LG1T and LG1-2T surpass them within a short horizon (Fig. 1 right). This early advantage stems from their optimistic exploration, which avoids purely random actions, whereas our methods begin with a brief uniform exploration phase. After fewer than 1000 steps, our algorithms consistently outperform all others over horizons of 20000 steps—despite the suboptimality of the 1-step and 2-step oracles.

While $\pi^{2,\text{greedy}}$ yields higher rewards than $\pi^{1,\text{greedy}}$, LG2T's slower convergence than LG1T (Thm.s 4.2 and 4.4) leads to slow initial reward accumulation. This is evident in Fig. 1, where LG1T outperforms LG2T over the early horizon (left) and over the full horizon in larger state spaces (right). As the state space grows, the convergence gap widens, and warm-starting becomes advantageous. LG1-2T is designed to balance rapid early convergence with better long-term performance. This hybrid achieves the best overall results in larger state spaces (Fig. 1, right). We also observe that model-based methods outperform model-free methods which aligns with observations from prior studies (Wei et al., 2020).

**JumpRiverSwim** We evaluate our algorithms on the JumpRiverSwim environment, a chain MDP with states $\{0, \cdots, S-1\}$, ($S = 5, 8, 15$), and actions $\{\texttt{left}, \texttt{right}\}$ where we discuss in detail in Ap-

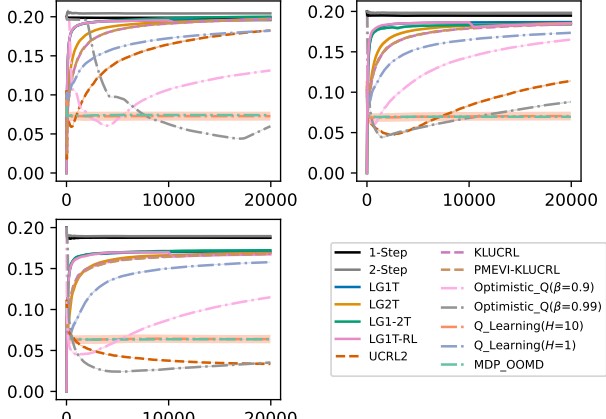

*Figure 2.* Running average reward under JumpRiverSwim environments averaged over 100 repetitions. Top Left: 5 states, Top Right: 8 states, Bottom Left: 15 states.

pendix E.1. In this setting, any K-step lookahead policy with short $K$ is suboptimal: it will myopically choose $\texttt{left}$ at state $0$, whereas only a sufficiently long-horizon lookahead discovers the optimal trajectory to the right. Despite this, our algorithms (LG1T, LG2T, and LG1-2T) consistently achieve higher cumulative reward than all RL baselines across all state-space sizes over 20,000 steps (Fig. 2).

The performance gain stems from our method's rapid identification of high-value actions at critical states. For example, LG1T quickly learns that $\texttt{right}$ is optimal at the rightmost state once that state is visited (which occurs early due to initial random exploration). In contrast, standard RL baselines must learn both the transition dynamics and the value function across the entire chain, leading to substantially slower convergence. Because no short-horizon (K-step) lookahead policy is optimal in this environment, we also implement a hybrid approach, LG1T-RL (Algorithm A.7), which switches from LG1T to the PMEVI-KLUCRL algorithm after 10,000 steps. As shown in Figure 2, LG1T-RL consistently matches or outperforms the standalone PMEVI-KLUCRL baseline and converges to identical behavior after the switch. This shows that our method can effectively warm-start a standard RL algorithm, delivering stronger finite-horizon performance during early learning while retaining asymptotic convergence to the optimal policy.

**FrozenLake** We evaluate our approach on the FrozenLake environment, a $4 \times 4$ grid world ($S = 16$) which we describe in detail in Appendix E.1. We omit oracle benchmarks as the underlying model is not directly accessible. As in JumpRiverSwim, an optimal policy requires end-of-horizon planning; any finite $K$ step lookahead oracle is suboptimal. Nevertheless, Figure 3 (Left) shows that our algorithms (LG1T, LG2T) consistently outperform all benchmark RL algorithms. This advantage arises because standard RL

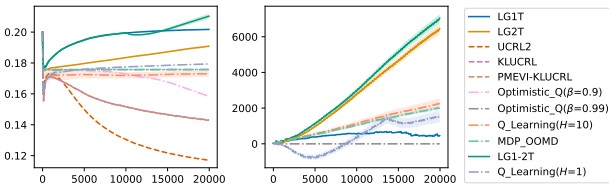

*Figure 3.* Left: Running average reward under FrozenLake environments with $4 \times 4$ state space averaged over 100 repetitions. LG1-2T has parameter $t_c = 10000$, Right: Cumulative reward of 20000 environment steps under AnyTrading environment averaged over 1000 repetitions. LG1-2T has parameter $t_c = 30$. Right exclude model based methods for prohibitive long runtime.

methods converge slowly and frequently fall into hazardous states, whereas our approach rapidly identifies and exploits low-risk trajectories, yielding higher cumulative reward.

### 5.2. Continuous State Space: AnyTrading

To address the scarcity of non-episodic benchmarks, we evaluate our algorithms in the AnyTrading environment, which simulates trading on real market data with continuous state spaces (stock prices) and binary actions (buy/sell). See Appendix E.1 for details. Given the computational cost of tabular model-based methods in continuous domains, we exclude them here. Experiments use environment resets with a fixed budget of 20,000 agent-environment interactions.

As shown in Figure 3 (Right), the suboptimality of the 1-step lookahead policy causes LG1T to be surpassed after a short horizon. In contrast, LG2T outperforms all other baselines within the budget. Moreover, warm-starting with LG1T in the LG1-2T variant yields even better performance, demonstrating the benefit of gradually increasing the planning horizon. A key insight from our experiments is that the fast convergence of our approach extends naturally from discrete to continuous state spaces, as demonstrated in the AnyTrading environment. This performance translates directly to superior results against benchmarks, even with raw state observations and no modification to the observation space. However, an important limitation arises from the tabular foundation of our methods: like all such RL approaches, they require multiple visits to similar states to identify optimal actions. This necessitates a degree of state recurrence—an assumption that may not hold in all continuous environments where states are rarely revisited. Extending this work to deep learning methods is an important direction for future research.

### 6. Conclusion

We conclude by explaining the empirical success of our method. As shown in Section 2.1, the linear lower bound on learning the optimal average-reward policy makes standard RL inherently slow in the non-episodic setting. Our approach circumvents this by targeting a K-step lookahead

thresholding policy, which enjoys a fast convergence rate. Notably, moving from a greedy to a thresholding rule is essential. While Q-learning with $H = 1$ (a UCB contextual bandit algorithm) also learns quickly in isolation, it can choose poor initial actions that trap the agent in low-value states, from which recovery is slow. Our thresholding mechanism uses lower confidence bounds to rapidly eliminate actions with low K-step reward, thereby accelerating learning and improving finite-horizon performance (Figs. 1–3). In our tested environments, neither $\pi^{1,\text{greedy}}$ nor $\pi^{2,\text{greedy}}$ is optimal. In the long run, standard RL algorithms would eventually surpass LG1T/LG2T. To retain both fast initial convergence and asymptotic optimality, we introduce a hybrid framework (Algorithm A.7) that transitions from LGKT to a standard RL method. As seen in Fig. 2, this retains the early boost without sacrificing eventual performance.

### Impact Statement

This paper presents work whose goal is to advance the field of Machine Learning. There are many potential societal consequences of our work, none which we feel must be specifically highlighted here.

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

# A. Additional Alorithms

---

**Algorithm A.3** Estimate-$r^{K-1}$

---

1: **Input:** sampling algorithm: $\text{ALG}_{K-1}$, reference state and action: $s_0, a_0$,initial state: $s'$ current step: $t$, cumulative reward 1- and K-step reward: $\hat{\phi}^1(s,a), \hat{\phi}^{K-1}(s,a), N_{s,a}^{(t)}, N_{s,a,K-1}^{(t)}$

2: Initialize: $s_t = s'$

3: **for** $k = t, t+1, t+K-2$ **do**

4:     Retrieve $s_k$, set $a_k = \text{ALG}_K(s_k | s, a)$

5:     Observe $(r_k, s_{k+1})$

6:     Update the cumulative one-step reward as follows:

7:         $\hat{\phi}^{\text{one}}(s_k, a_k) = \hat{\phi}^1(s_k, a_k) + r_k$

8:     Update the lower confidence bound of the K step reward as follows:

9:         $N_{s,a}^{(k+1)} = N_{s,a}^{(k)} + \mathbb{I}\{a_k = a, s_k = s\}, N_{s,a,K-1}^{(k+1)} = N_{s,a,K-1}^{(k)}$

10:        $\hat{r}_{s,a}^1(k+1) = \frac{\hat{\phi}^1(s,a)}{N_{s,a}^{(k+1)}}, \hat{r}_{s,a}^{K-1}(k+1) = \hat{r}_{s,a}^{K-1}(k)$

11:        $\text{LCB}_{s,a}^{(k+1)} = \hat{r}_{s,a}^1(k+1) + \hat{r}_{s,a}^{K-1}(k+1) - \sqrt{\frac{g\left(N_{s,a}^{(k+1)}+2\right)}{N_{s,a}^{(k+1)}+2}} - \sqrt{\frac{g\left(N_{s,a,K-1}^{(k+1)}+2\right)}{N_{s,a,K-1}^{(k+1)}+2}}$

12: **end for**

13: Update the cumulative $K-1$ step reward of $s_0, a_0$ as follows:

14:     $\hat{\phi}^{K-1}(s_0, a_0) = \hat{\phi}^{K-1}(s_0, a_0) + \sum_{k=t}^{t+K-2} r_k$

15: Update the lower confidence bound of the K step reward as follows:

16:     $N_{s_0,a_0,K-1}^{(t+K-1)} = N_{s_0,a_0,K-1}^{(t+K-1)} + 1$

17:     $\hat{r}_{s_0,a_0}^{K-1}(t+K-1) = \frac{\hat{\phi}^{K-1}(s_0,a_0)}{N_{s_0,a_0,K-1}^{(t+K-1)}}$

18:     $\text{LCB}_{s_0,a_0}^{(t+K-1)} = \hat{r}_{s_0,a_0}^1(t+K-1) + \hat{r}_{s_0,a_0}^{K-1}(t+K-1) - \sqrt{\frac{g\left(N_{s_0,a_0}^{(t+K-1)}+2\right)}{N_{s_0,a_0}^{(t+K-1)}+2}} - \sqrt{\frac{g\left(N_{s_0,a_0,K-1}^{(t+K-1)}+2\right)}{N_{s_0,a_0,K-1}^{(t+K-1)}+2}}$

19: **Return:** $N_{s,a,K-1}^{(t+K-1)}, N_{s,a}^{(t+K-1)}, \hat{\phi}^{K-1}, \hat{\phi}^1, \text{LCB}_{s,a}^{(t+K-1)}$

---

---

**Algorithm A.4** Update-LCB-1

---

**Input:** cumulative reward: $\hat{\phi}(s,a), N_{s,a}$, reward: $r$, state: $s_t$, action: $a_t$

Update the lower confidence bound of the reward as:

    $\hat{\phi}(s_t, a_t) = \hat{\phi}(s_t, a_t) + r$

    $N_{s,a} = N_{s,a} + \mathbb{I}\{a_t = a, s_t = s\}$

    $\hat{r}_{s,a}^1 = \hat{\phi}(s,a)/N_{s,a}$

    $\text{LCB}_{s,a} = \hat{r}_{s,a}^1 - \sqrt{\frac{g\left(N_{s,a}+2\right)}{N_{s,a}+2}}$

**Return:** $\hat{\phi}(s,a), N_{s,a}, \text{LCB}_{s,a}, \hat{r}_{s,a}^1$

---

---

**Algorithm A.5** Update-LCB-K

---

**Input:** cumulative 1-step lookahead reward: $\hat{\phi}^1_{s,a}$, cumulative K−1-step lookahead reward: $\hat{\phi}^{\mathrm{K}-1}_{s,a}$, $N_{s,a}, N_{s,a,\mathrm{K}-1}$, reward: $r$, state: $s_t$, action: $a_t$

Update the lower confidence bound of the K step reward as follows:

$\hat{\phi}^1_{s_t,a_t} = \hat{\phi}^1_{s_t,a_t} + r$

$N_{s,a} = N_{s,a} + \mathbb{I}\{a_t = a, s_t = s\}$

$\hat{r}^1_{s,a} = \frac{\hat{\phi}^1_{s,a}}{N_{s,a}}, \hat{r}^{\mathrm{K}-1}_{s,a} = \frac{\hat{\phi}^{\mathrm{K}-1}_{s,a}}{N_{s,a,\mathrm{K}-1}}$

$\mathrm{LCB}_{s,a} = \hat{r}^1_{s,a} + \hat{r}^{\mathrm{K}-1}_{s,a} - \sqrt{\frac{g(N_{s,a}+2)}{N_{s,a}+2}} - \sqrt{\frac{g(N_{s,a,\mathrm{K}-1}+2)}{N_{s,a,\mathrm{K}-1}+2}}$

**Return:** $\hat{\phi}^1(s,a), N_{s,a}, N_{s,a,\mathrm{K}-1}, \mathrm{LCB}_{s,a}, \hat{r}^1_{s,a}, \hat{r}^{\mathrm{K}-1}_{s,a}$

---

**Algorithm A.6** LCB-Guided 1-2 Step Thresholding (LG1-2T)

---

1: **Input:** initial state $s$, cutoff: $t_c$
2: **for** $t = 0, 1, \cdots, T$ **do**
3:   **if** $t \leq t_c$ **then**
4:     Run LG1T (Algorithm 1)
5:   **else**
6:     Run LG2T (Algorithm 2)
7:   **end if**
8: **end for**

---

**Algorithm A.7** LCB-Guided 1 Step Thresholding-RL (LG1T-RL)

---

1: **Input:** initial state $s$, cutoff: $t_c$
2: **for** $t = 0, 1, \cdots, T$ **do**
3:   **if** $t \leq t_c$ **then**
4:     Run LG1T (Algorithm 1)
5:   **else**
6:     Run PMEVI-KLUCRL (Boone & Zhang, 2024)
7:   **end if**
8: **end for**

---

## B. Technical Details of Section 3.1

### B.1. Proof of Theorem 3.3

We will first use induction to prove that $\boldsymbol{\pi}^{1,\mathrm{greedy}} = \boldsymbol{\pi}^*$. To ease notation, we will use $V^{\mathrm{greedy}}$ to denote $V^{\boldsymbol{\pi}^{1,\mathrm{greedy}}}$.

We will consider the following two cases: 1) $P_{a^*_{1,1}}(1,1) \geq P_{a^*_{0,1}}(0,1)$ and 2) $P_{a^*_{1,1}}(1,1) \leq P_{a^*_{0,1}}(0,1)$.

**Case 1:**   The induction hypothesis is:

$$\forall k \leq T, V^{\mathrm{greedy}}_k(1) \geq V^{\mathrm{greedy}}_k(0) \tag{6}$$

$$\pi^{1,\mathrm{greedy}}_k = \pi^*_k. \tag{7}$$

When $k = T$ this holds trivially, suppose it holds for $k \geq t+1$, when $k = t$: By optimal Bellman equation, we only need to prove the following to show Eq. (7):

$$r(s, a^*_{s,1}) + \sum_{s'} P_{a^*_{s,1}}(s, s') V^{\mathrm{greedy}}_{t+1}(s') \geq r(s, a) + \sum_{s'} P_a(s, s') V^{\mathrm{greedy}}_{t+1}(s').$$

We have

$$r(s, a^*_{s,1}) + \sum_{s'} P_{a^*_{s,1}}(s, s') V^{\mathrm{greedy}}_{t+1}(s') - r(s, a) - \sum_{s'} P_a(s, s') V^{\mathrm{greedy}}_{t+1}(s')$$

$$= r(s, a^*_{s,1}) - r(s, a) + \left( P_{a^*_{s,1}}(s, s') - P_a(s, s') \right) \left( V^{\text{greedy}}_{t+1}(1) - V^{\text{greedy}}_{t+1}(0) \right)$$

$$\geq 0,$$

where the last inequality holds because of Assumption 3.2 and induction hypothesis. This shows Eq. (7). For Eq. (6), we have

$$V^{\text{greedy}}_k(1) - V^{\text{greedy}}_k(0) = r(1, a^*_{1,1}) - r(0, a^*_{0,1}) + \left( P_{a^*_{1,1}}(1, 1) - P_{a^*_{0,1}}(0, 1) \right) \left( V^{\text{greedy}}_{k+1}(1) - V^{\text{greedy}}_{k+1}(0) \right)$$

$$\geq 0.$$

**Case 2:** The induction hypothesis is:

$$\forall k \leq T, 0 \leq V^{\text{greedy}}_k(1) - V^{\text{greedy}}_k(0) \leq r(1, a^*_{1,1}) - r(0, a^*_{0,1}) \tag{8}$$

$$\pi^{1,\text{greedy}}_k = \pi^*_k. \tag{9}$$

This holds trivially for $k = T$. Suppose it holds for $k \geq t + 1$, when $k = t$: Similar to Case 1, we can show Eq. (9) holds. We will first show the RHS of Eq. (8):

$$V^{\text{greedy}}_k(1) - V^{\text{greedy}}_k(0) = r(1, a^*_{1,1}) - r(0, a^*_{0,1}) + \left( P_{a^*_{1,1}}(1, 1) - P_{a^*_{0,1}}(0, 1) \right) \left( V^{\text{greedy}}_{k+1}(1) - V^{\text{greedy}}_{k+1}(0) \right)$$

$$\leq r(1, a^*_{1,1}) - r(0, a^*_{0,1}),$$

where the inequality holds because of induction hypothesis and $P_{a^*_{1,1}}(1, 1) - P_{a^*_{0,1}}(0, 1) \leq 0$. For LHS, we have

$$V^{\text{greedy}}_k(1) - V^{\text{greedy}}_k(0) = r(1, a^*_{1,1}) - r(0, a^*_{0,1}) + \left( P_{a^*_{1,1}}(1, 1) - P_{a^*_{0,1}}(0, 1) \right) \left( V^{\text{greedy}}_{k+1}(1) - V^{\text{greedy}}_{k+1}(0) \right)$$

$$\geq r(1, a^*_{1,1}) - r(0, a^*_{0,1}) + \left( P_{a^*_{1,1}}(1, 1) - P_{a^*_{0,1}}(0, 1) \right) (r(1, a^*_{1,1}) - r(0, a^*_{0,1}))$$

$$= \left( P_{a^*_{1,1}}(1, 1) + P_{a^*_{0,1}}(0, 0) \right) (r(1, a^*_{1,1}) - r(0, a^*_{0,1}))$$

$$\geq 0.$$

This proves $\pi^{1,\text{greedy}} = \pi^*$. Therefore, we have $\arg\max Q^*_k(s, a) = a^*_{s,1}$ for any $k \geq 0$ since $\pi^{1,\text{greedy}} = \pi^*$. By definition of $\pi^{\text{K,greedy}}$ (Eq. (3)), we have $\pi^{\text{K,greedy}}(a^*_{s,1}|s) = 1 = \pi^{1,\text{greedy}}(a^*_{s,1}|s)$. This shows that $\pi^{\text{K,greedy}} = \pi^{1,\text{greedy}} = \pi^*$. This concludes the proof.

### B.2. Proof of Theorem 3.4

We will let the state space be $\{B, G, D_1, D_2, \cdots, D_{S-2}\}$. And let the action space be $\{a_0, a_1, a_2, \cdots, a_{A-3}\}$. We will let $a_2, \cdots, a_{A-3}$ to have the same behaviour as $a_1$ and $D_1, D_2, \cdots, D_{S-2}$ have the same behaviour as $G$. We will define the transition matrix as:

$$P_{a_0}(B, B) = 1, P_{a_1}(B, G) = P_{a_1}(B, D_i) = 1/S - 1, P_{a_0}(G, G) = P_{a_0}(G, D_i) = 1/S - 1, P_{a_1}(G, B) = 1.$$

The reward $R$ is defined as:

$$R_{B,a_0} = -1, R_{B,a_1} = -(K + 1), R_{G,a_0} = 0, R_{G,a_1} = -1.$$

Therefore, it is easy to see that for $h \leq T - k + 1$, $Q^*_h(B, a_0) = T - h + 1, Q^*_h(B, a_1) = -(K + 1)$. Therefore, K-step lookahead thresholding policy will always choose $a_0$ at state $B$ when $\gamma$ is high. However, this will lead to $V^{\pi^{\text{K,greedy}}}_0(B) = -T - 1$ while $V^*_0(B) = -K - 1$. Therefore, the gap is $T - K$, which will grow linearly in $T$.

When threshold is low, i.e. $\gamma \leq -(K + 1)$, $\pi^{\text{K},\gamma}$ will become random policy, and it is easy to see that the gap also grows linearly in $T$.

## B.3. Proof of Theorem 3.5

We will use induction on the number of time until the end of horizon. Specifically, the hypothesis is

$$V_h^*(s) - V_h^{\boldsymbol{\pi}^{\mathrm{K},\boldsymbol{\gamma}}}(s) \le \sum_{t=h}^{T-K} \max_{s'} \left| P_{\pi_t^{\mathrm{K},\boldsymbol{\gamma}}}(s') - P_{a_{s',T-t+1}^*}(s') \right|_1 \max_{s'} V_{t+1}^*(s').$$

When $h \ge T - K + 1$, the hypothesis holds because the K-step lookahead policy is exactly the optimal policy. Suppose this holds when $h \ge l + 1$, when $h = l$,

$$V_l^*(s) - V_l^{\boldsymbol{\pi}^{\mathrm{K},\boldsymbol{\gamma}}}(s) = R(s, a_{s,T-l+1}^*) - \sum_a \pi_l^{\mathrm{K},\boldsymbol{\gamma}}(a|s) R(s,a)$$

$$+ \sum_{s'} P_{a_{s,T-l+1}^*}(s,s') V_{l+1}^*(s') - \sum_{s'} \sum_a \pi_l^{\mathrm{K},\boldsymbol{\gamma}}(a|s) P_a(s,s') V_{l+1}^{\boldsymbol{\pi}^{\mathrm{K},\boldsymbol{\gamma}}}(s')$$

$$\le \sum_{s'} \left( P_{a_{s,T-l+1}^*}(s,s') - \sum_a \pi_l^{\mathrm{K},\boldsymbol{\gamma}}(a|s) P_a(s,s') \right) V_{l+1}^*(s')$$

$$+ \sum_{s'} \sum_a \pi_l^{\mathrm{K},\boldsymbol{\gamma}}(a|s) P_a(s,s') \left( V_{l+1}^*(s') - V_{l+1}^{\boldsymbol{\pi}^{\mathrm{K},\boldsymbol{\gamma}}}(s') \right)$$

$$\le \left| \sum_{s'} \left( P_{a_{s,T-l+1}^*}(s,s') - \sum_a \pi_l^{\mathrm{K},\boldsymbol{\gamma}}(a|s) P_a(s,s') \right) \right| \max_{s'} \left| V_{l+1}^*(s') \right|$$

$$+ \max_{s'} V_{l+1}^*(s') - V_{l+1}^{\boldsymbol{\pi}^{\mathrm{K},\boldsymbol{\gamma}}}(s')$$

$$\le \sum_{t=l}^{T-K} \max_{s'} \left| P_{\pi_t^{\mathrm{K},\boldsymbol{\gamma}}}(s') - P_{a_{s',T-t+1}^*}(s') \right|_1 \max_{s'} V_{t+1}^*(s'),$$

where the last inequality holds because of the induction hypothesis. This completes the proof.

# C. Discussions Of Section 4

## C.1. Discussion On Assumption 4.3

When $K = 2$, then Assumption 4.3 is equivalent to assuming the regret of the contextual bandit problem is sublinear. Therefore, UCB algorithm satisfies the assumption with $C_{K-1} = A$ (Lattimore & Szepesvári, 2020). When $K > 2$, Assumption 4.3 relates with regret minimization in episodic RL with horizon length $K - 1$ and total number of episodes $H$. Specifically, episodic RL with horizon length $K - 1$ is maximizing $\mathbb{E}_{s_0 \sim P}\left[\sum_{k=0}^{K-2} R_{s_k,a_k} \mid s_0\right]$ within $H$ episodes. Further, the regret for episodic RL is defined as $\max \sum_{h=0}^{H-1} \mathbb{E}_{s_0 \sim P}\left[\sum_{k=0}^{K-2} R_{s_k,a_k}\right] - \sum_{k=0}^{K-2} R_{s_k^h,a_k^h}$ which mathces our assumption,. Therefore, algorithms such as UCBVI-BF (Azar et al., 2017) satisfies the assumption with $C_{K-1} = K^2 S A$.

We emphasized that though episodic RL algorithms with horizon length $K - 1$ can be used as the subroutine for Algorithm 2, it is not exclusive to episodic RL algorithms as any algorithms satisfying Assumption 4.3 is applicable. Further, Algorithm 2 is fundamentally different from episodic RL algorithm even when episodic RL algorithm is used as subroutine. As shown in Algorithm 2, the subroutine is merely used to calculate $K - 1$-step reward and will only be triggered in $\mathcal{O}(K\sqrt{T})$ times. Moreover, the goal of Algorithm 2 is choosing actions whose K-step lookahead reward is above the threshold while episodic RL algorithms will choose the action that maximizes the end-of-horizon lookahead reward. The fundamental difference in the learning goal makes our algorithm distinct from episodic RL algorithms.

# D. Technical Details of Section 4

Before proving Theorem , we first introduce Lemma D.1 which decomposes the cumulative regret to per-timestep regret.

**Lemma D.1.** *For any $t \le T$, let per-timestep regret $Regret_t(\pi, \mathcal{H}_t)$ for the policy $\boldsymbol{\pi}$ be defined as:*

$$Regret_t(\boldsymbol{\pi}, \mathcal{H}_t) = \mathbb{E}_{\pi_t}\left[c_t(a_t) \mid \mathcal{H}_t\right] + \mathbb{E}_{\pi_t}\left[\mathbb{E}_{\boldsymbol{\pi}^{1,\boldsymbol{\gamma}}}\left[\sum_{k=t+1}^{T} c_k(a_k) \mid r_t, s_{t+1}, a_t, \mathcal{H}_t\right]\right]$$

$$- \mathbb{E}_{\pi_t^1, \gamma} \left[ c_t\left(a_t\right) \mid \mathcal{H}_t \right] - \mathbb{E}_{\pi_t^1, \gamma} \left[ \mathbb{E}_{\boldsymbol{\pi}^1, \gamma} \left[ \sum_{k=t+1}^{T} c_k\left(a_k\right) \mid r_t, s_{t+1}, a_t, \mathcal{H}_t \right] \right].$$

*Then the cumulative regret $\mathcal{R}(\pi)$ can be decomposed as following:*

$$\mathcal{R}(\boldsymbol{\pi}) = \mathbb{E}\left[ \sum_{t=0}^{T} Regret_t\left(\boldsymbol{\pi}, \mathcal{H}_t\right) \right].$$

### D.1. Proof of Lemma D.1

*Proof.* We will prove the result iteratively from $t = 1$ to $T$. All we need is calculation. We have

$$\mathcal{R}(\boldsymbol{\pi}) = \mathbb{E}_{\boldsymbol{\pi}}\left[ \sum_{t=0}^{T} c_t(a_t) | s_0 \right] - \mathbb{E}_{\boldsymbol{\pi}^1, \gamma}\left[ \sum_{t=0}^{T} c_t(a_t) | s_0 \right]$$

$$= \mathbb{E}_{\pi_0}\left[ c_0(a_0) \mid \mathcal{H}_0 \right] + \mathbb{E}_{\pi_0}\left[ \mathbb{E}_{\boldsymbol{\pi}}\left[ \sum_{k=1}^{T} c_k\left(a_k\right) | r_0, s_1, a_1, \mathcal{H}_0 \right] \right]$$

$$- \mathbb{E}_{\pi_0}\left[ c_0(a_0) \mid \mathcal{H}_0 \right] - \mathbb{E}_{\pi_0}\left[ \mathbb{E}_{\boldsymbol{\pi}^1, \gamma}\left[ \sum_{k=1}^{T} c_k\left(a_k\right) | r_0, s_1, a_0, \mathcal{H}_0 \right] \right]$$

$$+ \mathbb{E}_{\pi_0}\left[ c_0(a_0) \mid \mathcal{H}_0 \right] + \mathbb{E}_{\pi_0}\left[ \mathbb{E}_{\boldsymbol{\pi}^1, \gamma}\left[ \sum_{k=1}^{T} c_k\left(a_k\right) | r_0, s_1, a_0, \mathcal{H}_0 \right] \right]$$

$$- \mathbb{E}_{\pi_0^1, \gamma}\left[ c_0\left(a_0\right) \mid \mathcal{H}_0 \right] - \mathbb{E}_{\pi_0^1, \gamma}\left[ \mathbb{E}_{\boldsymbol{\pi}^1, \gamma}\left[ \sum_{k=1}^{T} c_k\left(a_k\right) | r_0, s_1, a_0, \mathcal{H}_0 \right] \right]$$

$$= \mathbb{E}\left[ \mathbb{E}_{\boldsymbol{\pi}}\left[ \sum_{k=1}^{T} c_k\left(a_k\right) | \mathcal{H}_1 \right] - \mathbb{E}_{\boldsymbol{\pi}^1, \gamma}\left[ \sum_{k=1}^{T} c_k\left(a_k\right) | \mathcal{H}_1 \right] \right]$$

$$+ Regret\left(\boldsymbol{\pi}, \mathcal{H}_0\right).$$

We can continue this trick to $\mathbb{E}_{\boldsymbol{\pi}}\left[ \sum_{k=1}^{T} c_k\left(a_k\right) | \mathcal{H}_1 \right] - \mathbb{E}_{\boldsymbol{\pi}^1, \gamma}\left[ \sum_{k=1}^{T} c_k\left(a_k\right) | \mathcal{H}_1 \right]$, this will give us the desired result. $\qquad\square$

### D.2. Proof of Theorem 4.2

Before we prove the theorem, we first provide a technical lemma showing the concentration property of the empirical mean estimator $\hat{r}_{s,a}^{(t)}$, which is a direct result from Howard et al. 2021.

**Lemma D.2.** *For any $\alpha > 0$, $0 \le t \le T$, we have*

$$\mathbb{P}\left( \left| \hat{r}_{s,a}^{(t)} - r_{s,a}^{(t)} \right| \ge 1.7 \sqrt{\frac{\log \log (2 N_{s,m}(t)) + 0.72 \log(10.4/\alpha)}{N_{s,a}(t)}} \right) \le \alpha.$$

*Proof.* We will begin by the proof sketch: The proof proceeds in three steps 1) We show that analyzing the regret is equivalent to analyzing the time that we pull bad actions. 2) We show that pulling bad arms is caused by either overestimation of the reward for bad arms or underestimation of the reward for good arms. 3) We apply concentration inequality which does not require stationarity of the reward distribution (Lemma D.2). Moreover, with our design of lower confidence bound, we show that the time of overestimation is at most a constant.

The proof will proceed by first showing that the cumulative regret can be decomposed into per-timestep regret $Regret_t(\boldsymbol{\pi}, \mathcal{H}_t)$. By Lemma D.1, we only need to bound $Regret_t(\pi, \mathcal{H}_t)$. We have

$$Regret_t(\pi, \mathcal{H}_t) = \mathbb{E}_{\pi_t}\left[ c_t(a_t) \mid \mathcal{H}_t \right] + \mathbb{E}_{\pi_t}\left[ \mathbb{E}_{\boldsymbol{\pi}^1, \gamma}\left[ \sum_{k=t+1}^{T} c_k\left(a_k\right) | r_t, a_t, s_{t+1}, \mathcal{H}_t \right] \right]$$

$$- \mathbb{E}_{\pi_t^{1,\gamma}} \left[ c_t \left( a_t \right) \mid \mathcal{H}_t \right] - \mathbb{E}_{\pi_t^{1,\gamma}} \left[ \mathbb{E}_{\boldsymbol{\pi}^{1,\gamma}} \left[ \sum_{k=t+1}^{T} c_k \left( a_k \right) \mid r_t, s_{t+1}, a_t, \mathcal{H}_t \right] \right]$$

$$= \mathbb{E}_{\pi_t} \left[ c_t(a_t) \mid \mathcal{H}_t \right] + \mathbb{E}_{\pi_t} \left[ \mathbb{E}_{\boldsymbol{\pi}^{1,\gamma}} \left[ \sum_{k=t+1}^{T} c_k \left( a_k \right) \mid r_t, s_{t+1}, a_t, \mathcal{H}_t \right] \right]$$

$$\leq \mathbb{E} \left[ \sum_{\Delta_{s,a}^1 > 0} \Delta_{s,a}^1 \mathbb{I} \left\{ a_t = a, s_t = s \right\} \right] + \mathbb{E}_{\pi_{t+1}} \left[ V_{t+1}^{\boldsymbol{\pi}^{1,\gamma}} \left( s_{t+1} \right) \mid s_{t+1}, a_t, \mathcal{H}_t \right]$$

$$= \mathbb{E} \left[ \sum_{\Delta_{s,a}^1 > 0} \Delta_{s,a}^1 \mathbb{I} \left\{ a_t = a, s_t = s \right\} \right],$$

where the second and the third equality holds by Assumption 4.1. Then we have the cumulative regret is upper bounded by

$$\sum_{\Delta_{s,a}^1 > 0} \Delta_{s,a}^K \mathbb{E} \left[ \sum_{t=0}^{T} \mathbb{I} \left\{ a_t = a, s_t = s \right\} \right].$$

We have

$$\mathbb{E} \left[ \sum_{t=0}^{T} \mathbb{I} \left\{ a_t = a, s_t = s \right\} \right] = \mathbb{E} \left[ \sum_{t=0}^{T} \mathbb{I} \left\{ a_t = a, s_t = s, \text{LCB}_{s,a}^{(t)} \geq \gamma \right\} \right] \tag{10}$$

$$+ \mathbb{E} \left[ \sum_{t=0}^{T} \mathbb{I} \left\{ a_t = a, s_t = s, \text{LCB}_{s,a}^{(t)} \leq \gamma \right\} \right]. \tag{11}$$

We will bound the two terms separately. For Eq. (10), we have

$$\mathbb{E} \left[ \sum_{t=0}^{T} \mathbb{I} \left\{ a_t = a, s_t = s, \text{LCB}_{s,a}^{(t)} \geq \gamma \right\} \right]$$

$$= \mathbb{E} \left[ \sum_{k=1}^{\infty} \mathbb{I} \left\{ \text{LCB}_{s,a}^{(\tau_{s,a}^k - 1)} \geq \gamma, \tau_{s,a}^k \leq T \right\} \right]$$

$$\leq \mathbb{E} \left[ \sum_{k=1}^{T} \mathbb{I} \left\{ \hat{r}_{s,a}^1 (\tau_{s,a}^k - 1) \geq \gamma + \sqrt{\frac{3 \log \left( N_{s,a}^{(\tau_{s,a}^k)} \right)}{N_{s,a}^{(\tau_{s,a}^k)}}} \right\} \right]$$

$$\leq \sum_{k=1}^{T} \exp \left( -N_{s,a}^{(\tau_{s,a}^k)} \left( \Delta_{s,a}^1 \right)^2 + \log \log \left( N_{s,a}^{(\tau_{s,a}^k)} \right) - 3 \log \left( N_{s,a}^{(\tau_{s,a}^k)} \right) \right)$$

$$\leq \sum_{k=1}^{T} \frac{1}{k^2} \exp \left( -k \left( \Delta_{s,a}^1 \right)^2 \right)$$

$$\leq 2, \tag{12}$$

where $\tau_{s,a}^k := \inf \{ t : N_{s,a}^{(t)} = k \}$ and the second inequality holds because of Lemma D.2.

For Eq. (11), we have

$$\mathbb{E} \left[ \sum_{t=0}^{T} \mathbb{I} \left\{ a_t = a, s_t = s, \text{LCB}_{s,a}^{(t)} \leq \gamma \right\} \right]$$

$$= \mathbb{E} \left[ \sum_{t=0}^{T} \mathbb{I} \left\{ a_t = a, s_t = s, \text{LCB}_{s,a}^{(t)} \leq \gamma, \exists a' \in \mathcal{G}_s, \text{LCB}_{s,a'}^{(t)} \leq \gamma \right\} \right]$$

$$\leq \mathbb{E}\left[\sum_{t=0}^{T}\sum_{a'\in\mathcal{G}_s}\mathbb{I}\left\{a_t = a, s_t = s, \mathrm{LCB}_{s,a}^{(t)} \leq \gamma, \mathrm{LCB}_{s,a'}^{(t)} \leq \gamma\right\}\right]$$

$$= \sum_{a'\in\mathcal{G}_s}\mathbb{E}\left[\sum_{t=0}^{T}\mathbb{I}\left\{a_t = a, s_t = s, \mathrm{LCB}_{s,a}^{(t)} \leq \gamma, \mathrm{LCB}_{s,a'}^{(t)} \leq \gamma\right\}\right]$$

For each $a' \in \mathcal{G}_s$, we have

$$\mathbb{E}\left[\sum_{t=0}^{T}\mathbb{I}\left\{a_t = a, s_t = s, \mathrm{LCB}_{s,a}^{(t)} \leq \gamma, \mathrm{LCB}_{s,a'}^{(t)} \leq \gamma\right\}\right] = \mathbb{E}\left[\sum_{t=0}^{T}\mathbb{I}\left\{a_t = a', s_t = s, \mathrm{LCB}_{s,a}^{(t)} \leq \gamma, \mathrm{LCB}_{s,a'}^{(t)} \leq \gamma\right\}\right]$$

$$\leq \mathbb{E}\left[\sum_{t=0}^{T}\mathbb{I}\left\{a_t = a', s_t = s, \mathrm{LCB}_{s,a'}^{(t)} \leq \gamma\right\}\right]$$

$$= \mathbb{E}\left[\sum_{k=1}^{\infty}\mathbb{I}\left\{\mathrm{LCB}_{s,a'}^{(\tau_{s,a'}^k - 1)} \geq \gamma, \tau_{s,a'}^k \leq T\right\}\right]$$

$$\leq \mathbb{E}\left[\sum_{k=1}^{T}\mathbb{I}\left\{\hat{r}_{s,a'}^1(\tau_{s,a'}^k - 1) \geq \gamma + \sqrt{\frac{3\log\left(N_{s,a'}^{\left(\tau_{s,a'}^k\right)}\right)}{N_{s,a'}^{\left(\tau_{s,a'}^k\right)}}}\right\}\right]$$

$$\leq u + \mathbb{E}\left[\sum_{k=u+1}^{T}\mathbb{I}\left\{\hat{r}_{s,a'}^1(\tau_{s,a'}^k - 1) \geq \gamma + \sqrt{\frac{3\log\left(N_{s,a'}^{\left(\tau_{s,a'}^k\right)}\right)}{N_{s,a'}^{\left(\tau_{s,a'}^k\right)}}}\right\}\right]$$

$$\leq u + \sum_{k=u+1}^{T}\frac{1}{k^2}$$

$$\leq u + 2,$$

where $u = \frac{12}{\left(\Delta_+^1\right)^2}\left(\log\left(\frac{e}{\left(\Delta_+^1\right)^6}\right) + \log\log\left(\frac{1}{\left(\Delta_+^1\right)^6}\right)\right)$. We should note that such choice of $u$ satisfies $\sqrt{3\log(u)/u} \leq \Delta_+^1$. Therefore, Eq. (18) can be upper bounded as

$$\mathbb{E}\left[\sum_{t=0}^{T}\mathbb{I}\left\{a_t = a, s_t = s, \mathrm{LCB}_{s,a}^{(t)} \leq \gamma\right\}\right]$$

$$= \sum_{a'\in\mathcal{G}_s}\mathbb{E}\left[\sum_{t=0}^{T}\mathbb{I}\left\{a_t = a, s_t = s, \mathrm{LCB}_{s,a}^{(t)} \leq \gamma, \mathrm{LCB}_{s,a'}^{(t)} \leq \gamma\right\}\right]$$

$$\leq A(u+2). \tag{13}$$

Combining Eq. (12) and Eq. (13) will give us the desired result.

Further assume that $C_{\mathrm{diff}}\left|\Delta_+^1\right| \geq \Delta_{s,a}^1$, for all $\Delta_{s,a}^1 > 0$. Then we have

$$\mathrm{R}(\pi) \leq \sum_{\Delta_{s,a}^1 \leq \Delta}\Delta_{s,a}^1\mathbb{E}\left[T_{s,a}\right] + \sum_{\Delta_{s,a}^1 \geq \Delta}\frac{24C_{\mathrm{diff}}}{\Delta_+^1}\log\left(\frac{e}{\left(\Delta_+^1\right)^6}\right) + \sum_{\Delta_{s,a}^1 > 0}(2A+2)\Delta_{s,a}^1$$

$$\leq \Delta T + SA24C_{\mathrm{diff}}\frac{1}{\Delta}\log\left(\frac{e}{\Delta^6}\right) + \sum_{\Delta_{s,a}^1 > 0}(2A+2)\Delta_{s,a}^1.$$

Take $\Delta = \sqrt{\sqrt{\frac{24C_{\text{diff}}SA}{T}}}$ will give us the desired result. □

### D.3. Proof of Theorem 4.4

*Proof.* The same as proof of Theorem 4.2, the proof will proceed by first showing that the cumulative regret can be decomposed into per-timestep regret $\text{Regret}_t(\pi, \mathcal{H}_t)$. Then we will use the standard technique to bound the number of times suboptimal actions are given. We make key modifications: 1) *Regret from exploration*: choosing $\epsilon_t = \mathcal{O}(\sqrt{N^{(t)}_{s_{t-1},a_{t-1}}})$ ensures that the cumulative regret due to entering the exploration phase is at most $K\sqrt{SAT}$; 2) *Sufficient estimation samples*: the same choice of $\epsilon_t$ ensures that for each state-action pair $s, a$, at least $\frac{K}{2}\sqrt{SAN^{(T)}_{s,a}}$ times picking actions to estimate $r^{\text{K}}_{s,a}$. Recall that the K-step lookahead reward comprises a 1-step lookahead reward and a K−1-step lookahead reward. Accurate estimation of the latter is ensured by Assumption 4.3. We show that $\frac{K}{2}\sqrt{SAN^{(T)}_{s,a}}$ runs of $\text{ALG}_{K-1}$ suffices to obtain an accurate estimation of $r^{\text{K}}$.

Using the same method, we have the cumulative regret is upper bounded by

$$\sum_{\Delta^{\text{K}}_{s,a}>0} \Delta^{\text{K}}_{s,a}\mathbb{E}\left[\sum_{t=0}^{T}\mathbb{I}\{a_t = a, s_t = s\}\right].$$

We have

$$\sum_{\Delta^{\text{K}}_{s,a}>0} \Delta^{\text{K}}_{s,a}\mathbb{E}\left[\sum_{t=0}^{T}\mathbb{I}\{a_t = a, s_t = s\}\right] = \sum_{\Delta^{\text{K}}_{s,a}>0} \Delta^{\text{K}}_{s,a}\mathbb{E}\left[\sum_{t=0}^{T}\mathbb{I}\{a_t = a, s_t = s, \text{(K-1)-step}\}\right] \tag{14}$$

$$+ \sum_{\Delta^{\text{K}}_{s,a}>0} \Delta^{\text{K}}_{s,a}\mathbb{E}\left[\sum_{t=0}^{T}\mathbb{I}\{a_t = a, s_t = s, \text{K-step}\}\right]. \tag{15}$$

We will first bound Eq. (14):

$$\sum_{\Delta^{\text{K}}_{s,a}>0} \Delta^{\text{K}}_{s,a}\mathbb{E}\left[\sum_{t=0}^{T}\mathbb{I}\{a_t = a, s_t = s, \text{(K-1)-step}\}\right] = \sum_{t=0}^{T}\mathbb{E}\left[\sum_{\Delta^{\text{K}}_{s,a}>0} \Delta^{\text{K}}_{s,a}\mathbb{I}\{a_t = a, s_t = s, \text{(K-1)-step}\}\right]$$

$$\leq \sum_{t=0}^{T} C_{\text{diff}}\Delta^{\text{K}}\mathbb{E}\left[\mathbb{I}\{\text{(K-1)-step}\}\right]$$

$$\leq (K-1)\sum_{t=0}^{T} C_{\text{diff}}\Delta^{\text{K}}\frac{1}{\Delta^{\text{K-1}}\sqrt{N^{(t)}_{s_{t-1},a_{t-1}}}}$$

$$\leq (K-1)C_{\text{diff}}\sum_{s,a}\sqrt{N^{(T)}_{s,a}}$$

$$\leq (K-1)C_{\text{diff}}\sqrt{SAT}, \tag{16}$$

where the last inequality holds because of Cauchy-Schwarz.

Then we will bound Eq. (15) We have

$$\mathbb{E}\left[\sum_{t=0}^{T}\mathbb{I}\{a_t = a, s_t = s, \text{K-step}\}\right] = \mathbb{E}\left[\sum_{t=0}^{T}\mathbb{I}\left\{a_t = a, s_t = s, \text{LCB}^{(t)}_{s,a} \geq \gamma\right\}\right] \tag{17}$$

$$+ \mathbb{E}\left[\sum_{t=0}^{T}\mathbb{I}\left\{a_t = a, s_t = s, \text{LCB}^{(t)}_{s,a} \leq \gamma\right\}\right]. \tag{18}$$

We will bound the two terms separately. For Eq. (17), we have

$$\mathbb{E}\left[\sum_{t=0}^{T}\mathbb{I}\left\{a_t = a, s_t = s, \text{LCB}_{s,a}^{(t)} \geq \gamma\right\}\right]$$

$$= \mathbb{E}\left[\sum_{k=1}^{\infty}\mathbb{I}\left\{\text{LCB}_{s,a}^{(\tau_{s,a}^k - 1)} \geq \gamma, \tau_{s,a}^k \leq T\right\}\right]$$

$$\leq \mathbb{E}\left[\sum_{k=1}^{T}\mathbb{I}\left\{\text{LCB}_{s,a}^{(\tau_{s,a}^k - 1)} \geq \gamma, N_{s,a,\text{K}-1}^{(\tau_{s,a}^k - 1)} \geq \max\left\{\frac{\sqrt{k-1}}{2\Delta^{\text{K}}}, 2\sqrt{k-1}\right\}\right\}\right]$$

$$+ \mathbb{E}\left[\sum_{k=1}^{T}\mathbb{I}\left\{N_{s,a,\text{K}-1}^{(\tau_{s,a}^k - 1)} \leq \max\left\{\frac{\sqrt{k-1}}{2\Delta^{\text{K}}}, 2\sqrt{k-1}\right\}\right\}\right] \tag{19}$$

where $\tau_{s,a}^k := \inf\{t : N_{s,a}^{(t)} = k\}$. Since $N_{s,a,\text{K}-1}^{(\tau_{s,a}^k - 1)} = \sum_{l=1}^{k-1}\mathbb{I}\left\{\epsilon \leq \frac{1}{\min\{\Delta^{\text{K}}, 1/2\}\sqrt{l}}\right\}$, where $\epsilon \sim \text{Uniform}(0,1)$, we have the second term is upper bounded by $\frac{8}{(\Delta^{\text{K}})^2}$ because of Chernoff bound. Next, we will bound the first term of Eq. 19.

$$\mathbb{E}\left[\sum_{k=1}^{T}\mathbb{I}\left\{\text{LCB}_{s,a}^{(\tau_{s,a}^k - 1)} \geq \gamma, N_{s,a,\text{K}-1}^{(\tau_{s,a}^k - 1)} \geq \max\left\{\frac{\sqrt{k-1}}{2\Delta^{\text{K}}}, 2\sqrt{k-1}\right\}\right\}\right]$$

$$= \mathbb{E}\left[\sum_{k=1}^{T}\mathbb{I}\left\{\text{LCB}_{s,a}^{(\tau_{s,a}^k - 1)} \geq \gamma, N_{s,a,\text{K}-1}^{(\tau_{s,a}^k - 1)} \geq \max\left\{\frac{\sqrt{k-1}}{2\Delta^{\text{K}}}, 2\sqrt{k-1}\right\}, \left|\mathbb{E}\left[\hat{r}_{s,a}^{\text{K}-1}(\tau_{s,a}^k - 1)\right] - r_{s,a_{\text{K}-1,s}^*}^{\text{K}-1}\right| \leq \sqrt{\frac{C_{K-1}\log(T)}{N_{s,a,\text{K}-1}^{(\tau_{s,a}^k - 1)}}}\right\}\right] \tag{20}$$

$$+ \mathbb{E}\left[\sum_{k=1}^{T}\mathbb{I}\left\{\left|\mathbb{E}\left[\hat{r}_{s,a}^{\text{K}-1}(\tau_{s,a}^k - 1)\right] - r_{s,a_{\text{K}-1,s}^*}^{\text{K}-1}\right| \geq \sqrt{\frac{C_{K-1}\log(T)}{N_{s,a,\text{K}-1}^{(\tau_{s,a}^k - 1)}}}\right\}\right]. \tag{21}$$

By Assumption 4.3, Eq. (21) is smaller than $\mathbb{E}\left[\sum_{k=1}^{T} 1/T\right] = 1$. To ease notation, we will use $\mathcal{E}_A^k$ to denote $N_{s,a,\text{K}-1}^{(\tau_{s,a}^k - 1)} \geq \max\left\{\frac{\sqrt{k-1}}{2\Delta^{\text{K}}}, 2\sqrt{k-1}\right\}$ and $\mathcal{E}_B^k$ to denote $\left|\mathbb{E}\left[\hat{r}_{s,a}^{\text{K}-1}(\tau_{s,a}^k - 1)\right] - r_{s,a_{\text{K}-1,s}^*}^{\text{K}-1}\right| \leq \sqrt{\frac{C_{K-1}\log(T)}{N_{s,a,\text{K}-1}^{(\tau_{s,a}^k - 1)}}}$. For Eq. (20), we have

$$\mathbb{E}\left[\sum_{k=1}^{T}\mathbb{I}\left\{\text{LCB}_{s,a}^{(\tau_{s,a}^k - 1)} \geq \gamma, \mathcal{E}_A^k, \mathcal{E}_B^k\right\}\right] \tag{22}$$

$$= \mathbb{E}\left[\sum_{k=1}^{T}\mathbb{I}\left\{\text{LCB}_{s,a}^{(\tau_{s,a}^k - 1)} - r_{s,a} - \mathbb{E}\left[\hat{r}_{s,a}^{\text{K}-1}(\tau_{s,a}^k - 1)\right] \geq \gamma - r_{s,a} - r_{s,a_{\text{K}-1,s}^*}^{\text{K}-1} + r_{s,a_{\text{K}-1,s}^*}^{\text{K}-1} - \mathbb{E}\left[\hat{r}_{s,a}^{\text{K}-1}(\tau_{s,a}^k - 1)\right], \mathcal{E}_A^k, \mathcal{E}_B^k\right\}\right]$$

$$\leq \frac{64C_{K-1}^2\log(T)}{(\Delta^{\text{K}})^2} + \mathbb{E}\left[\sum_{k=64C_{K-1}^2\log(T)/(\Delta^{\text{K}})^2}^{T}\mathbb{I}\left\{\text{LCB}_{s,a}^{(\tau_{s,a}^k - 1)} - r_{s,a} - \mathbb{E}\left[\hat{r}_{s,a}^{\text{K}-1}(\tau_{s,a}^k - 1)\right] \geq \frac{\Delta^{\text{K}}}{2}, \mathcal{E}_A^k\right\}\right]$$

$$\leq \frac{64C_{K-1}^2\log(T)}{(\Delta^{\text{K}})^2} + \mathbb{E}\left[\sum_{k=1}^{T}\mathbb{I}\left\{\hat{r}_{s,a}^1(\tau_{s,a}^k - 1) - r_{s,a} \geq \frac{\Delta^{\text{K}}}{4} + \sqrt{\frac{3\log(N_{s,a}^{(\tau_{s,a}^k - 1)})}{N_{s,a}^{(\tau_{s,a}^k - 1)}}}\right\}\right]$$

$$+ \mathbb{E}\left[\sum_{k=1}^{T}\mathbb{I}\left\{\hat{r}_{s,a}^{\text{K}-1}(\tau_{s,a}^k - 1) - \mathbb{E}\left[\hat{r}_{s,a}^{\text{K}-1}(\tau_{s,a}^k - 1)\right] \geq \frac{\Delta^{\text{K}}}{4} + \sqrt{\frac{3\log(N_{s,a,\text{K}-1}^{(\tau_{s,a}^k - 1)})}{N_{s,a,\text{K}-1}^{(\tau_{s,a}^k - 1)}}}, \mathcal{E}_A^k\right\}\right]$$

$$\leq \frac{64C_{K-1}^2\log(T)}{(\Delta^{\text{K}})^2} + \sum_{k=1}^{T}\exp\left(-N_{s,a}^{(\tau_{s,a}^k)}\left(\Delta^{\text{K}}\right)^2 + \log\log\left(N_{s,a}^{(\tau_{s,a}^k)}\right) - 3\log\left(N_{s,a}^{(\tau_{s,a}^k)}\right)\right)$$

$$+ \exp\left(-2\sqrt{k}\left(\Delta^{\text{K}}\right)^2 + \log\log\left(2\sqrt{k}\right) - 3\log\left(2\sqrt{k}\right)\right)$$

$$\leq \frac{64 C_{K-1}^2 \log(T)}{(\Delta^{\mathrm{K}})^2} + \sum_{k=1}^{T} \frac{1}{k^2} \exp\left(-k \left(\Delta^{\mathrm{K}}\right)^2\right) + \frac{1}{2k}$$

$$\leq \frac{128 C_{K-1}^2 \log(T)}{(\Delta^{\mathrm{K}})^2}, \tag{23}$$

where the thrid last inequality holds because of Lemma D.2.

For Eq. (18), we have

$$\mathbb{E}\left[\sum_{t=0}^{T} \mathbb{I}\left\{a_t = a, s_t = s, \mathrm{LCB}_{s,a}^{(t)} \leq \gamma\right\}\right]$$

$$= \mathbb{E}\left[\sum_{t=0}^{T} \mathbb{I}\left\{a_t = a, s_t = s, \mathrm{LCB}_{s,a}^{(t)} \leq \gamma, \exists (a') \in \mathcal{G}_f, \mathrm{LCB}_{s,a'}^{(t)} \leq \gamma\right\}\right]$$

$$\leq \sum_{a' \in \mathcal{G}_s} \mathbb{E}\left[\sum_{t=0}^{T} \mathbb{I}\left\{a_t = a, s_t = s, \mathrm{LCB}_{s,a}^{(t)} \leq \gamma, \mathrm{LCB}_{s,a'}^{(t)} \leq \gamma\right\}\right]$$

$$= \sum_{a' \in \mathcal{G}_s} \mathbb{E}\left[\sum_{t=0}^{T} \mathbb{I}\left\{a_t = a', s_t = s, \mathrm{LCB}_{s,a}^{(t)} \leq \gamma, \mathrm{LCB}_{s,a'}^{(t)} \leq \gamma\right\}\right]$$

$$\leq \sum_{a' \in \mathcal{G}_s} \mathbb{E}\left[\sum_{t=0}^{T} \mathbb{I}\left\{a_t = a', s_t = s, \mathrm{LCB}_{s,a'}^{(t)} \leq \gamma\right\}\right]$$

For each $a' \in \mathcal{G}_s$, we have

$$\mathbb{E}\left[\sum_{t=0}^{T} \mathbb{I}\left\{a_t = a', s_t = s, \mathrm{LCB}_{s,a'}^{(t)} \leq \gamma\right\}\right] \leq \mathbb{E}\left[\sum_{k=1}^{T} \mathbb{I}\left\{\mathrm{LCB}_{s,a'}^{(\tau_{s,a'}^k - 1)} \leq \gamma, \mathcal{E}_A^k, \mathcal{E}_B^k\right\}\right] + \frac{8}{(\Delta^{\mathrm{K}})^2}, \tag{24}$$

where the inequality holds similar to bounding Eq. (17).

$$\mathbb{E}\left[\sum_{k=1}^{T} \mathbb{I}\left\{a_t = a', s_t = s, \mathrm{LCB}_{s,a'}^{(\tau_{s,a'}^k)} \leq \gamma, \mathcal{E}_A^k, \mathcal{E}_B^k\right\}\right]$$

$$= \mathbb{E}\left[\sum_{k=1}^{T} \mathbb{I}\left\{\mathrm{LCB}_{s,a'}^{(\tau_{s,a'}^k - 1)} - r_{s,a'} - \mathbb{E}\left[\hat{r}_{s,a}^{\mathrm{K}-1}(\tau_{s,a'}^k - 1)\right] \leq \gamma - r_{s,a'} - r_{s,a_{\mathrm{K}-1,s}^*}^{\mathrm{K}-1} + r_{s,a_{\mathrm{K}-1,s}^*}^{\mathrm{K}-1} - \mathbb{E}\left[\hat{r}_{s,a'}^{\mathrm{K}-1}(\tau_{s,a'}^k - 1)\right], \mathcal{E}_A^k, \mathcal{E}_B^k\right\}\right]$$

$$\leq u + \mathbb{E}\left[\sum_{k=u+1}^{T} \mathbb{I}\left\{\mathrm{LCB}_{s,a'}^{(\tau_{s,a'}^k - 1)} - r_{s,a'} - \mathbb{E}\left[\hat{r}_{s,a'}^{\mathrm{K}-1}(\tau_{s,a'}^k - 1)\right] \leq \frac{\Delta^{\mathrm{K}}}{2}, \mathcal{E}_A^k\right\}\right]$$

$$\leq u + \mathbb{E}\left[\sum_{k=1}^{T} \mathbb{I}\left\{\hat{r}_{s,a'}^1(\tau_{s,a'}^k - 1) - r_{s,a'} \leq -\frac{\Delta^{\mathrm{K}}}{4} + \sqrt{\frac{3\log(N_{s,a'}^{(\tau_{s,a'}^k - 1)})}{N_{s,a'}^{(\tau_{s,a'}^k - 1)}}}\right\}\right]$$

$$+ \mathbb{E}\left[\sum_{k=1}^{T} \mathbb{I}\left\{\hat{r}_{s,a'}^{\mathrm{K}-1}(\tau_{s,a'}^k - 1) - \mathbb{E}\left[\hat{r}_{s,a'}^{\mathrm{K}-1}(\tau_{s,a'}^k - 1)\right] \leq -\frac{\Delta^{\mathrm{K}}}{4} + \sqrt{\frac{3\log(N_{s,a',\mathrm{K}-1}^{(\tau_{s,a'}^k - 1)})}{N_{s,a',\mathrm{K}-1}^{(\tau_{s,a'}^k - 1)}}}, \mathcal{E}_A^k\right\}\right]$$

$$\leq u + \sum_{k=1}^{T} \exp\left(-N_{s,a'}^{(\tau_{s,a'}^k)} \left(\Delta^{\mathrm{K}}\right)^2 + \log\log\left(N_{s,a'}^{(\tau_{s,a'}^k)}\right) - 3\log\left(N_{s,a'}^{(\tau_{s,a'}^k)}\right)\right)$$

$$+ \exp\left(-2\sqrt{k}\left(\Delta^{\mathrm{K}}\right)^2 + \log\log\left(2\sqrt{k}\right) - 3\log\left(2\sqrt{k}\right)\right)$$

$$\leq u + \sum_{k=1}^{T} \frac{1}{k^2} \exp\left(-k \left(\Delta^{\mathrm{K}}\right)^2\right) + \frac{1}{2k}$$

$$\leq 2u, \tag{25}$$

where $u = \max\left\{ \frac{64 C_{K-1}^2 \log(T)}{(\Delta^{\mathrm{K}})^2}, \frac{144}{(\Delta^{\mathrm{K}})^2} \left( \log\left(\frac{e}{(\Delta^{\mathrm{K}})^6}\right) + \log\log\left(\frac{1}{(\Delta^{\mathrm{K}})^6}\right) \right) \right\}$. We should note that such choice of $u$ satisfies $\sqrt{3 \log(u)/u} \leq \Delta^{\mathrm{K}}/4$ and $\sqrt{2 \Delta^{\mathrm{K}} C_{K-1} \log(T)}/\sqrt{(u-1)} \leq \Delta^{\mathrm{K}}/2$.

Combining Eq. (16), Eq. (23), Eq. (24) and Eq. (25), we have the regret is upper bounded by $(K-1) C_{\mathrm{diff}} \sqrt{SAT} + \frac{256 C_{\mathrm{diff}} C_{k-1}^2 SA \log(T)}{\Delta^{\mathrm{K}}}$. Then we have $\mathcal{R} \leq \min\{(K-1) C_{\mathrm{diff}} \sqrt{SAT}, 16 C_{\mathrm{diff}} C_{K-1} \sqrt{SAT \log(T)}\}$ $\qquad \square$

# E. Experiment Details

## E.1. Environment Details

**Synthetic MDP:** We randomly generate 1000 instances with 10 states and 5 actions and generate 1000 instances with 100 states and 25 actions. The mean reward $R(s,a)$ is generated according to $\mathrm{Gamma}(0.5, 1)$ and the reward is normal with mean $R(s,a)$ and variance 0.5. For 10 state case, the transition probabilities are generated according to $\mathrm{Gamma}(0.1, 10)$ and then normalized. For 100 state case, the transition probabilities are generated according to $\mathrm{Gamma}(0.01, 1000)$.

**JumpRiverSwim (Wei et al., 2020):** JumpRiverSwim models a swimmer who can choose to swim left or right in a river. The states are arranged in a chain and the swimmer starts from the leftmost state $s = 0$. We will denote the rightmost state as $S$. At state 0, if the swimmer chooses to swim left, they will go to any arbitrary state with probability 0.01 and they will stay at state 0 with probability $1 - 0.01$. If the swimmer chooses to swim right, they will go to any arbitrary state with probability 0.01, stay at state 0 with probability $0.7 + 0.01/(S+1)$ and go to the right state with probability $0.3 - 0.01$. At state 1, if the swimmer chooses to swim right, they will go to any arbitrary state with probability 0.01, go to the left state with probability $0.7 + 0.01/(S+1)$ and stay at $S$ with probability $0.3 - 0.01$. If the swimmer chooses to swim left, they will go to any arbitrary state with probability 0.01 and they will go to the left state with probability $1 - 0.01$. At any other states, if the swimmer chooses to swim left, they will go to any arbitrary state with probability 0.01 and they will go to the left state with probability $1 - 0.01$. If they choose to swim right, they will go to any arbitrary state with probability 0.01, go to the left state with probability $0.6 + 0.01/(S+1)$, go to the right state with probability $0.3 - 0.01$ and stay at current state with probability $0.1 + 0.01/S$. The reward is all zero except at state 0 and state $S$. Specifically $r(0,0) = 0.2, r(S,1) = 1$.

In our experiment, we set $S$ to be $4, 7, 14$.

**FrozenLake (Brockman et al., 2016):** The game starts at position $[0, 0]$ and the goal is to reach the end state for as many times as possible. The goal is at far extent of the game environment. At each time, the player will choose to go to one of its four neighbors. If the player is at a frozen tile, it may go along perpendicular to the intended direction with probability $1/3$. If the player reaches hole or goal, it will immediately go back to the starting state. In order to encourage avoiding holes, we set the reward when in hole be 0, the reward when on frozen tile to be 0.2 and the reward when reaches the goal to be 1. We test on a $4 \times 4$ grid with the following two maps.

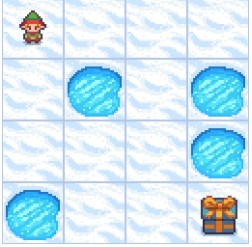

*Figure 4.* The FrozenLake environment.

**AnyTrading (Haghpanah et al., 2023):** This is a simulated trading environment built from real dataset. The state space is an array that contains the close price of the previous day an thus is continuous. It contained two actions: sell or buy. We use the default FOREX environment contained in Haghpanah et al. (2023).

## E.2. Implementation Details

Since to the best our knowledge, no existing algorithms directly address non-episodic finite horizon MDP, we choose algorithms for infinite horizon average reward and episodic RL as benchmarks. We chose algorithms that are theoretically minimax optimal (Boone & Zhang, 2024) and have been reported empirically to work well (Wei et al., 2020; Jin et al., 2018).

Specifically, we used the following benchmarks:

- UCRL2 (Auer et al., 2008): At each step, construct a confidence ball of transition probabilities using $\mathcal{L}_1$ distance. Then they will use extended value iteration (EVI) to calculate the policy that maximizes the average reward of MDP in the confidence region.

- KLUCRL (Filippi et al., 2010): At each step, construct a confidence ball of transition probabilities using KL divergence. Then they will use extended value iteration to calculate the policy that maximizes the average reward of MDP in the confidence region.

- PMEVI-KLUCRL (Boone & Zhang, 2024): This replaces the extended value iteration with projection mitigated extended value iteration which solves extended value iteration with a span constraint. They use the same way to construct confidence ball as KLUCRL.

- Optimistic Q Learning (Wei et al., 2020): This learns a Q function of discounted reward MDP with high discount factor and chooses greedily according to the Q function. As suggested in the paper, we make two choices of discount factor $\gamma$: 0.9 and 0.99. We note that since this learns discounted reward Q function, it can also be seen as discounted reward algorithml.

- MDP-OOMD (Wei et al., 2020): This applies policy mirror decent to average reward setting.

- Q Learning (Jin et al., 2018): This learns the Q function for finite horizon MDP in episodic setting. In order to maximize the number of episodes, we let the horizon of the Q function be $1, 10$.

The implementation of the model based methods (Auer et al., 2008; Filippi et al., 2010; Boone & Zhang, 2024) and model free average reward methods (Wei et al., 2020) are the same as the original implementation and the hyperparameters are the same as the ones suggested in those papers.

## E.3. Additional Experiments on Adaptively Choosing $\gamma$

In this section, we provide a heuristic of changing the threshold: starting from the lower bound of the K-step lookahead reward and then increase the threshold by $\Delta$ for state $s$ when the number of visit $N_s^{(t)}$ exceeds $\log(T)$. As shown in Figures 5 and 6, this heuristic (LG1T-I) allows us to start LG1T from initial threshold being 0 for all states while the overall performance could exceed the performance of starting at higher threshold on synthetic environments and FrozenLake. Same behviour happens to LG2T This makes sure that the algorithm can learn fast at the beginning and begin to learn a better policy when there is sufficient knowledge. We did not compare on other environments since Figures 15-18 shows different threshold does not make big difference.

## E.4. Additional Experiments on Adaptively Changing K

In this section, we provide a heuristic of the data-driven change time of Algorithm A.6. Instead of choosing a fixed change time, we will let it be $\sqrt{SAT}$. This matches the regret bound for LG1T and serve as a sign for learning 1-step lookahead policy well. As shown in Figures 7-9 , this heuristic (LG1-2T-Adaptive) outperforms LG1T on all environments and outperform LG2T except the 10 state synthetic instances.

## E.5. Additional Experiments on Theory Version of Algorithms 1, 2

As shown in Figures 10-12, LG1T-Uniform with the same or smaller threshold has similar performance compared with LG1T that uses UCB when no actions' LCB is above the threshold and consistently beats the baselines on all environments we tested. LG2T-Uniform with the same or smaller threshold also has similar performance compared with LG2T on all

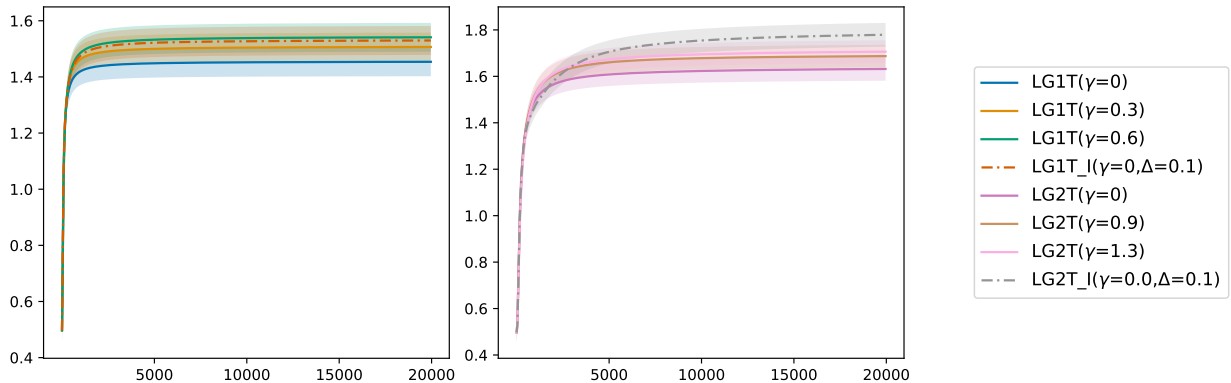

*Figure 5.* Comparison of adaptive choosing threshold on 10 states and 5 actions synthetic environment. Right: LG1T. Left: LG2T.

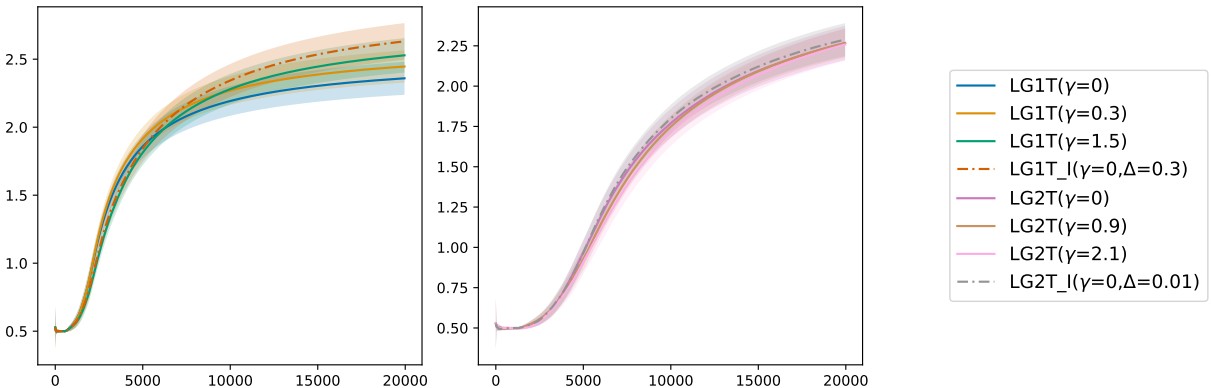

*Figure 6.* Comparison of adaptive choosing threshold on 100 states and 25 actions synthetic environment. Right: LG1T. Left: LG2T.

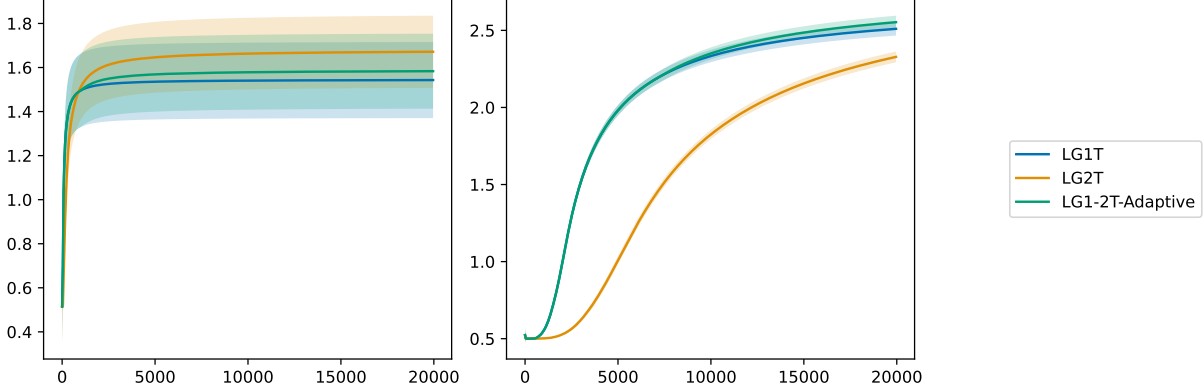

*Figure 7.* Comparison of adaptive choosing the changing time. Right: 10 states and 5 actions. Left: 100 states and 25 actions.

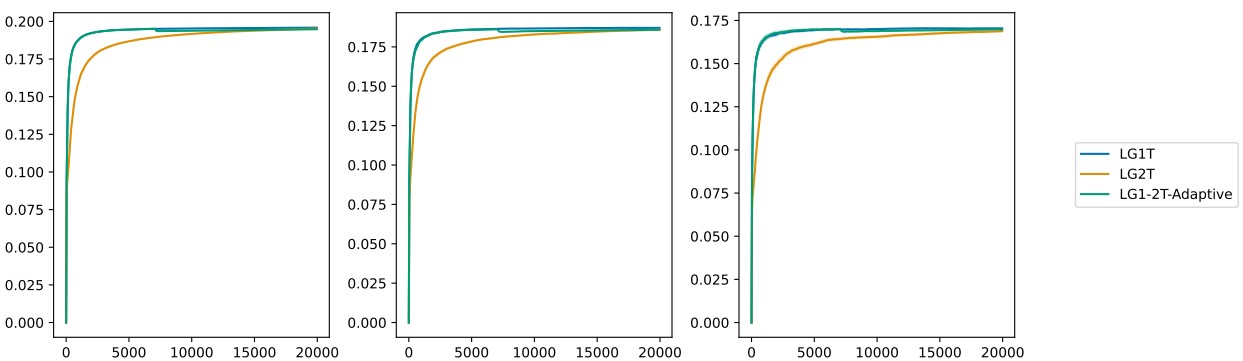

*Figure 8.* Comparison of adaptive choosing the changing time on JumpRiverswim. Right: 5 states. Middle: 8 states. Left: 15 states.

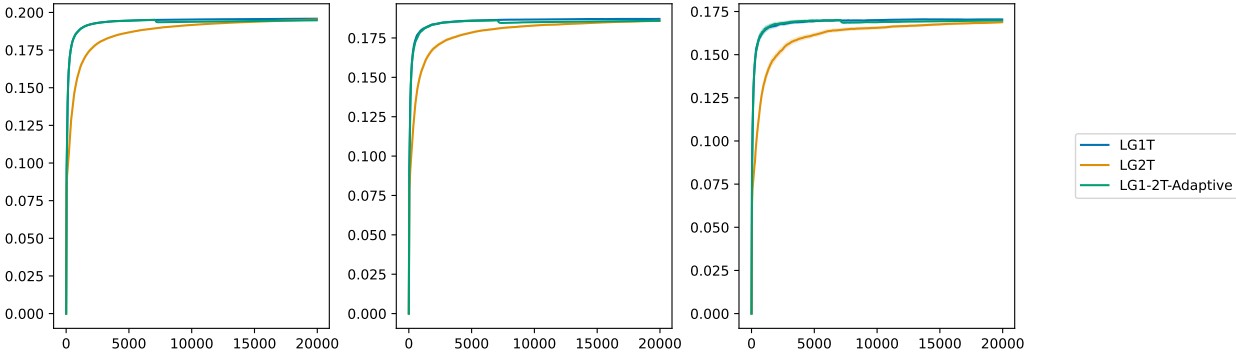

*Figure 9.* Comparison of adaptive choosing the changing time on FrozenLake.

environments. On JumpRiverswim environment, LG2T is worse than PMEVI-KLUCRL, this supports our point: when all actions' LCB is below the threshold, which can happen since the reward of many middle state is 0, using UCB can give us a better way of choosing the action with potentially higher reward.

These results indicate the theory version of Algorithms 1, 2 can also be the best among all benchmarks but the implemented version can further improve the theory version.

### E.6. Ablation on $\gamma$

In this section, we will test the performance of our algorithms for a various choices of threshold on all instances we have tested. As shown in Figures 13-17, a high threshold will lead to better asymptotic performance which matches our observation that with higher threshold, $\pi^{K,\gamma}$ will have better performance. However, we also observe that in the synthetic MDP with $S = 100, A = 25$, higher threshold will first be worse than lower threshold in the beginning because as we mentioned in Section 4, higher threshold will slower the convergence. Nevertheless, as shown in the figures, the difference is small and this shows that in practice, a moderate choice of threshold can be chosen to balance between maximizing the convergence and maximizing the reward. Notably, the specific threshold used in our main experiments (Section 5), while not tuned for peak performance, still enables our algorithms to consistently outperform all benchmarks, underscoring the robustness of our design.

## F. Regret Bounds for Implemented Algorithm 1

**Theorem F.1.** *Under Assumption 4.1 with* $K = 1$*, the regret* $\mathcal{R}^{\pi^{1,\gamma}}$ *of implemented Algorithm 1 satisfies* $\mathcal{R}^{\pi^{1,\gamma}} = \mathcal{O}\left(\sqrt{SAT\log(T)}\right).$

*Proof.* Since the only change is when no arm's LCB is higher than the threshold, we only need to bound Eq. (11). The other

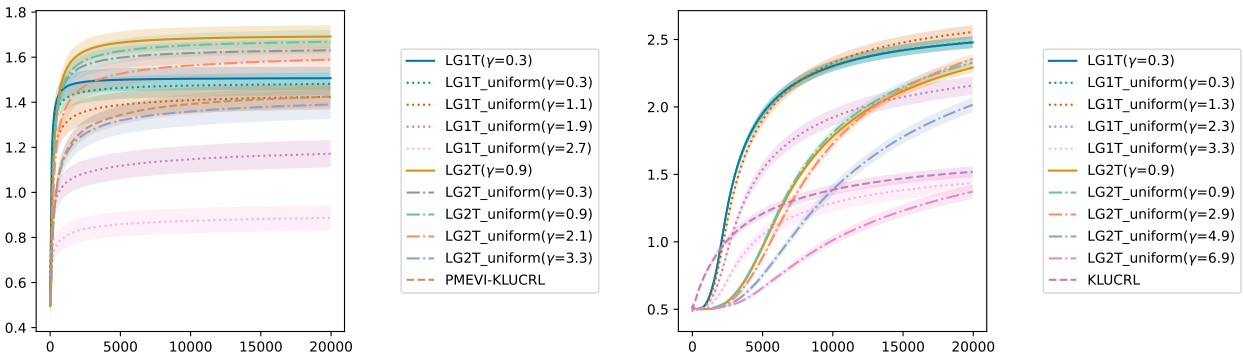

*Figure 10.* Comparison of theory version and implemented version on 1000 synthetic MDPs. Right: 10 states and 5 actions. Left: 100 states and 25 actions.

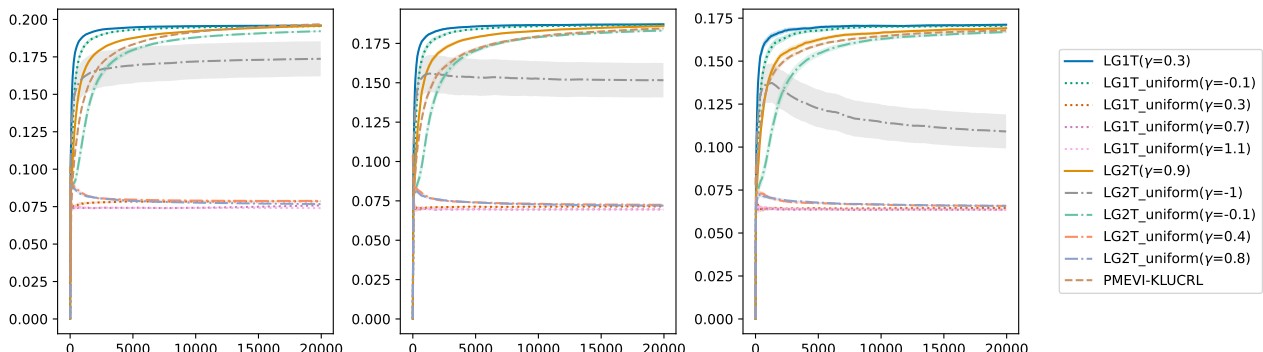

*Figure 11.* Comparison of theory version and implemented version on JumpRiverswim. Right: 5 states. Middle: 8 states. Left: 15 states.

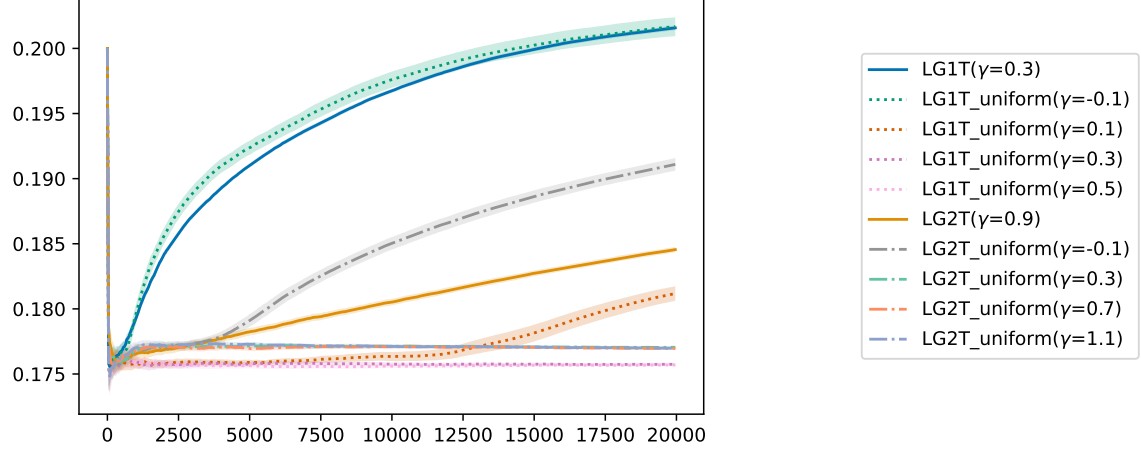

*Figure 12.* Comparison of theory version and implemented version on FrozenLake.

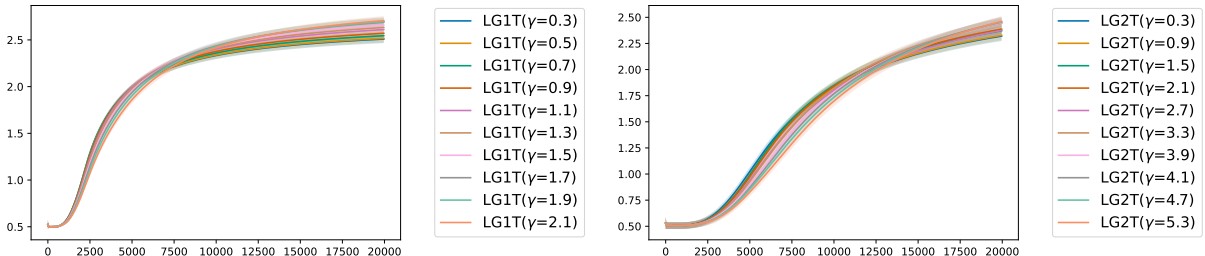

*Figure 13.* Ablations on 1000 synthetic MDPs. $S = 100, A = 25$

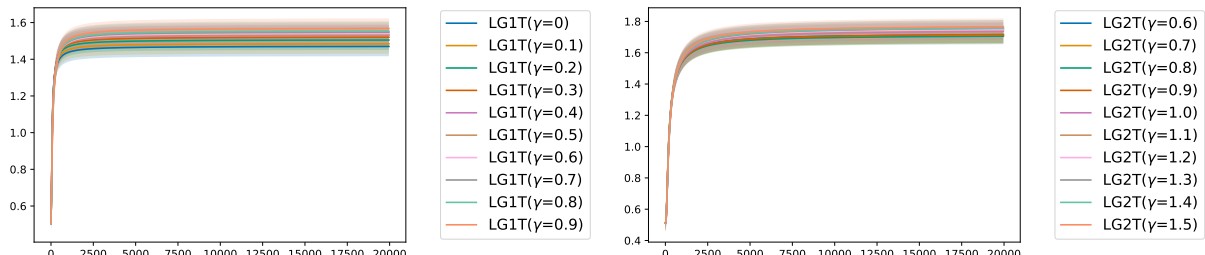

*Figure 14.* Ablations on 1000 synthetic MDPs. $S = 10, A = 5$

term will be the same as the proof of Theorem 4.2. We have

$$
\mathbb{E}\left[\sum_{t=0}^{T} \mathbb{I}\left\{a_t = a, s_t = s, \mathrm{LCB}_{s,a}^{(t)} \leq \gamma\right\}\right]
$$

$$
= \mathbb{E}\left[\sum_{t=0}^{T} \mathbb{I}\left\{a_t = a, s_t = s, \mathrm{LCB}_{s,a}^{(t)} \leq \gamma, \mathrm{UCB}_{s,a}^{(t)} \geq \mathrm{UCB}_{s,a_{s,1}^*}^{(t)}\right\}\right]
$$

$$
\leq \mathbb{E}\left[\sum_{t=0}^{T} \mathbb{I}\left\{s_t = s, \mathrm{UCB}_{s,a}^{(t)} \geq \mathrm{UCB}_{s,a_{s,1}^*}^{(t)}\right\}\right]
$$

$$
\leq \sqrt{A N_s^{(T)} \log(T)},
$$

where the last inequality holds using the same technique as in (Lattimore & Szepesvári, 2020). Then we have the regret is upper bounded by $\sum_s \sqrt{A N_s^{(T)} \log(T)} \leq \sqrt{SAT \log(T)}$. This completes the proof. $\qquad\square$

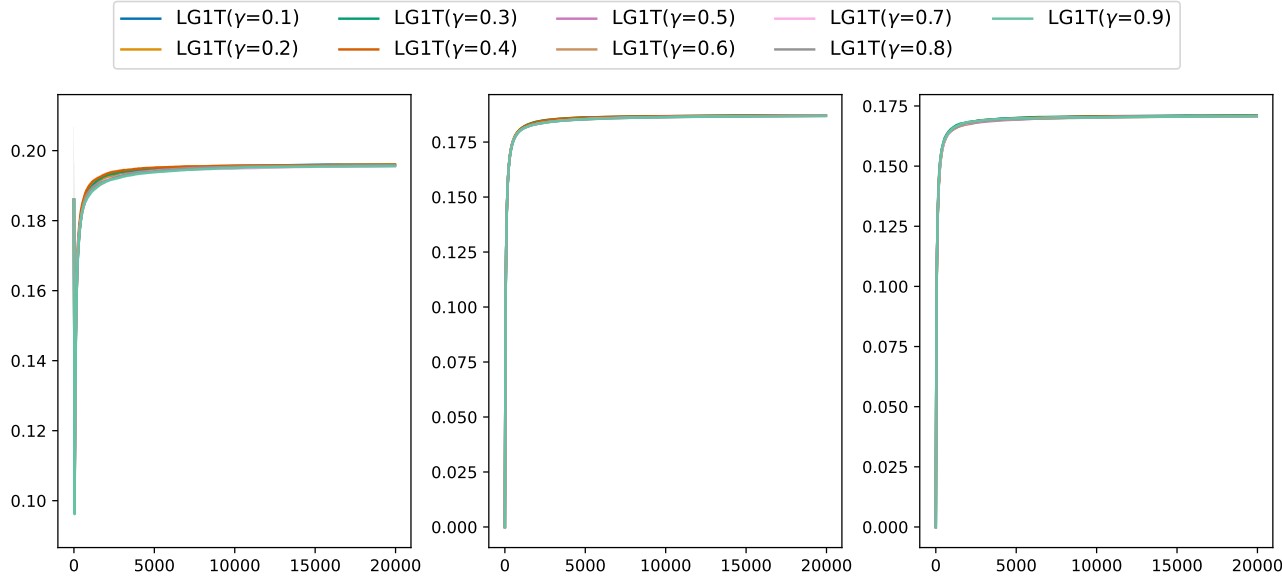

*Figure 15.* Ablations on JumpRiverSwims. Left: $S = 5$, Middle: $S = 8$, Right: $S = 15$.

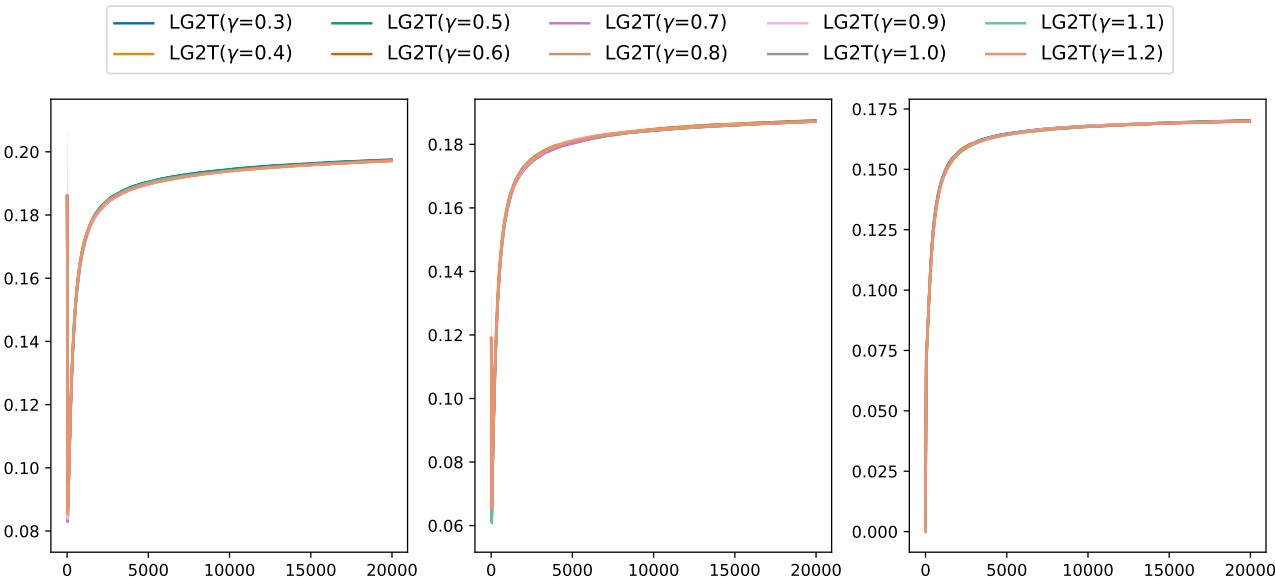

*Figure 16.* Ablations of LG2T on JumpRiverSwims. Left: $S = 5$, Middle: $S = 8$, Right: $S = 15$.

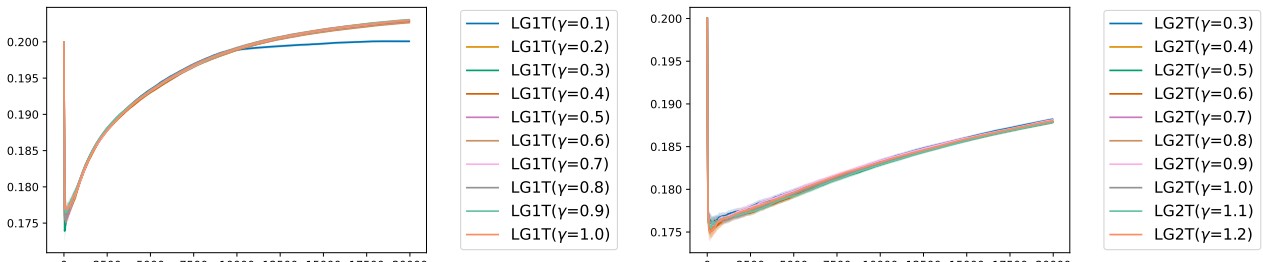

*Figure 17.* Ablations on FrozenLake. Left: LG1T, Right: LG2T.

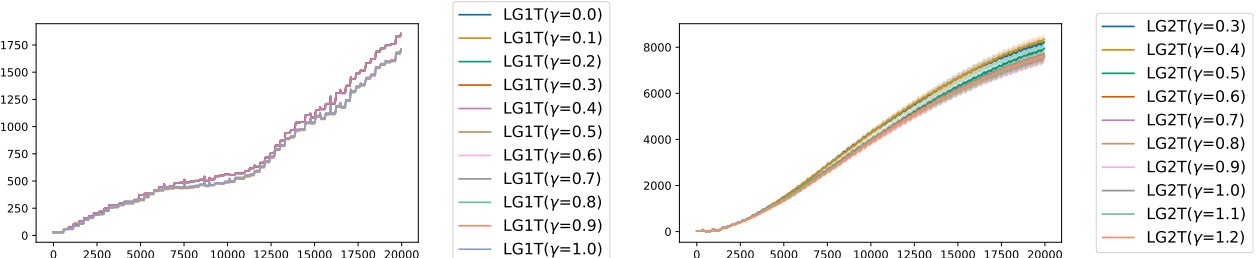

*Figure 18.* Ablations on AnyTrading. Left: LG1T, Right: LG2T.

