# OpenReview forum: "Fast Non-Episodic Finite-Horizon RL with K-Step Lookahead Thresholding"
_ICML.cc/2026/Conference — ICML 2026 regular_

### Official Review · Reviewer_D7tf · 2026-03-06

**Soundness:** 3
**Presentation:** 2
**Significance:** 3
**Originality:** 3
**Overall Recommendation:** 5
**Confidence:** 4

**Summary:**

This paper studies reinforcement learning in finite-horizon MDPs without episodic resets. Instead of optimizing the full finite-horizon value function, the authors propose learning truncated $K$-step Q-values and identifying actions whose truncated return exceeds a fixed threshold $\gamma$. Based on this idea, they introduce two algorithms: LG1T, which handles the $K=1$ case (reducing the problem to a thresholding contextual bandit), and LGKT, which generalizes the approach to arbitrary $K$ by estimating truncated $K$-step returns via rollouts. The paper provides theoretical guarantees showing that these algorithms identify the corresponding $K$-lookahead thresholding policy and establishes regret bounds, including constant regret in the $K=1$ case and guarantees under additional assumptions for $K>1$.

**Compliance With Llm Reviewing Policy:**

Affirmed.

**Final Justification:**

The paper is solid, and the authors have addressed my main concerns. The idea is both interesting and well motivated, suggesting the use of shorter-horizon objectives as a stepping stone toward more complex problems. Overall, this is a clear accept to me.

**Key Questions For Authors:**

* How does the threshold policy $\pi^{\gamma,K}$ relate to the optimal policy $\pi^\star$? While the regret guarantees are stated with respect to the threshold policy, it would be helpful to better understand how suboptimal this policy can be and whether there are conditions under which $\pi^{\gamma,K}$ is guaranteed to be close to optimal.

* For the algorithms with $K>1$, the estimation of truncated returns requires $K$-step rollouts starting from a state–action pair $(s,a)$. How are these rollouts conducted in practice in the non-episodic setting? In particular, does the method assume access to environment resets, a generative model, or some mechanism to revisit specific states?

* The motivation focuses on non-episodic finite-horizon tasks where only a single trajectory may be observed. However, the experiments involve long interaction horizons with many simulation steps. In which scenarios do we expect the non-episodic formulation to be particularly relevant, and how does the method differ in practice from standard episodic RL when sufficient interaction data is available?

**Limitations:**

Yes

**Strengths And Weaknesses:**

**Strengths:** Regarding strong points of the paper:

* The problem considered is interesting and relatively underexplored. Non-episodic finite-horizon MDPs have received limited attention in the literature, despite being challenging settings where the learner observes a single trajectory rather than repeated episodes.
* The paper explains the problem setting clearly and gradually builds the ideas. The progression from the simplest case ($K=1$), which reduces to a bandit problem, to more complex cases where $K$ increases is easy to follow.
* The paper provides theoretical analysis supporting the proposed methods, including guarantees for the $K=1$ case and extensions to larger $K$, which helps illustrate the potential benefits of the approach.

**Weak points:** The paper shows the following limitations:

* The regret analysis is carried out with respect to a thresholding policy that selects actions whose truncated value exceeds a threshold $\gamma$. However, it is unclear how suboptimal this policy can be relative to the optimal policy. While Theorem 4 provides some characterization for the $K$-greedy policy, the relationship between that policy and the thresholding policy actually analyzed in the regret results is not clearly established.
* The analysis relies on the assumption of a “good sampling policy” capable of estimating truncated $K$-step returns. In practice this requires generating rollouts of length $K$ from a given state–action pair $(s,a)$. It is not entirely clear how this can be achieved in an online non-episodic setting without access to resets or a generative model.
* The motivation focuses on finite-horizon problems without episodic resets. However, when the horizon is large the setting becomes effectively close to the infinite-horizon case, and the distinction between episodic and non-episodic interaction becomes less relevant. The most interesting regime is therefore short horizons where only a single trajectory is observed, but the experiments do not clearly reflect this scenario. In particular, the empirical evaluation is conducted over long interaction horizons (e.g., on the order of tens of thousands of steps), which effectively provides the learner with extensive data and repeated state visits. As a result, the experiments do not clearly capture the regime where the lack of episodic resets and the availability of only a short trajectory would make the proposed approach particularly necessary.

---

> ### Author Rebuttal · Authors · 2026-03-30
>
> We thank the reviewer for the positive assessment, particularly for recognizing the importance of the non-episodic finite-horizon setting, the clarity from the K=1 bandit case to general K, and the supporting theory. We address the concerns below.
>
> **Suboptimality Gap:** As noted in Line 181, the greedy policy is a special case of thresholding under a high threshold. Theorem 3.4 shows its worst-case gap is linear, and the same holds for K-step thresholding; we will add this proof.
>
> We will also add an instance-dependent bound: $V_0^{\pi^{optimal}}(s) - V_0^{\pi^{K,greedy}}\leq\sum_{t=0}^{T-K-1}\max_s\|\sum_a \pi^{\gamma,K}(a|s) P_{s,a}-P_{s,a_{a,T-t}^*}\|_1\max_s V_0^{\pi^{optimal}}(s)$
> The proof is using backward induction. This bound characterizes that the constant of the linear depends on the $L_1$ distance between the transition probability under action that maximizes the K-step reward and the optimal action. Therefore, when the two transition probability is similar, K-step thresholding policy is near optimal. We note that, although the current bound may not be tight, deriving a sharper and more meaningful characterization of the suboptimality gap is substantially harder in our setting. In discounted MDPs, such analyses typically exploit the contraction property of the Bellman operator. This key tool is unavailable in the undiscounted finite-horizon setting considered here. As a result, a tighter characterization would require directly analyzing the transition dynamics induced by the
> K-step thresholding policy relative to those of the optimal policy, which is technically much more challenging. Our focus in this paper is instead on the learnability side: we ask whether one can design an online algorithm that converges quickly to the
> K-step thresholding policy, and whether this faster convergence can translate into higher cumulative reward. Empirically, this is exactly what we observe: across all tested environments, LG2T consistently outperforms all baselines despite the potential suboptimality gap. We therefore believe that tightly characterizing this gap and identifying the corresponding favorable regimes is an important direction for future work, but beyond the scope of the present paper.
>
> **Rollout of Subroutine:** Our method does *not* require rollouts from arbitrary state–action pairs, resets, or a generative model. The subroutine is only invoked on *observed* trajectory data. When triggered at time $t$, the rollout starts from $(s_{t-1}, a_{t-1})$ and proceeds along the trajectory.
>
>  Thus, the “good sampling policy” assumption ensures informative trajectory-based rollouts, not simulator access. We will clarify this in Appendix C.2.
>
> **Long Horizon:** Plots show average reward up to $t$; early parts already reflect the limited-data regime when each policy only interacts with the environment $t$ times. Longer horizons we plot are intended to demonstrate the persistence of fast convergence gains.
>
> The key distinction between our method and episodic RL algorithms is the learning target: we learn a $K$-step surrogate that can be identified faster, while standard RL targets the optimal policy and may converge slowly. This distinction is particularly relevant in finite-horizon settings where the convergence rate to the optimal policy is linear. Therefore, converging rapidly to a strong surrogate may lead to a higher cumulative reward than slowly approaching the exact optimum. We will add a concluding paragraph discussing future extension in the revised version:
>
> Our idea of learning a K-step thresholding policy, which has faster convergence to enable better performance when the horizon is short, can potentially extend to cases when we have an infinite horizon or large episodes where there are many samples. We view combining learning K-step thresholding policy with RL algorithms that aim to learn an optimal policy to have both faster convergence at the beginning and asymptotically optimality as important future work.

---

> > ### Author Rebuttal · Reviewer_D7tf · 2026-04-01
> >
> > I thank the authors for the clarifying response, particularly for the bound that more clearly relates the greedy and thresholding policies. While the bound may not be tight, it provides useful intuition about the relationship between both policies.
> >
> > The discussion on the experimental setup is also relevant. Finite-horizon problems can serve as surrogate targets to learn near-optimal policies efficiently, with potential transfer to the infinite-horizon setting, which represents a meaningful extension of the work.
> >
> > Overall, I am satisfied with the rebuttal and will therefore increase my score.

---

> > > ### Author Response · Authors · 2026-04-01
> > >
> > > Thank you for raising the scores and your thoughtful comments!

---

### Official Review · Reviewer_KRaG · 2026-03-08

**Soundness:** 2
**Presentation:** 2
**Significance:** 3
**Originality:** 2
**Overall Recommendation:** 4
**Confidence:** 4

**Summary:**

This paper studies online reinforcement learning in non-episodic finite-horizon MDPs and proposes to learn a surrogate target policy rather than directly optimizing the full-horizon optimum. The surrogate target is defined through K-step lookahead with thresholding: instead of identifying the exact best action under a longer planning horizon, the learner aims to identify actions that are sufficiently good according to a thresholded criterion. The paper argues that this target can be learned substantially faster, producing better cumulative reward under finite interaction budgets even though it may be suboptimal relative to the true full-horizon optimum. Theoretical results characterize regret relative to this surrogate target, and experiments compare against several tabular RL baselines on synthetic and benchmark environments.

**Compliance With Llm Reviewing Policy:**

Affirmed.

**Final Justification:**

The rebuttal addressed my concerns

**Key Questions For Authors:**

1. Can the authors more explicitly characterize the regime in which the proposed surrogate target is expected to outperform full-horizon-oriented alternatives under finite budget? For example, are there problem properties under which the target-quality / learnability trade-off is especially favorable?

2. Can the authors provide an ablation or discussion comparing the theoretically faithful LCB-style version with the UCB-based heuristic used in experiments? This would help clarify whether the empirical gains come from the thresholding principle itself or from additional optimistic exploration effects introduced in implementation.

3. How sensitive is the method to reduced state revisitation? If possible, it would be helpful to include experiments or discussion on environments where states are revisited less often, to better delineate the practical scope of the approach.

**Limitations:**

See weakness

**Strengths And Weaknesses:**

Strengths:

1. The paper studies a meaningful and relatively uncommon RL setting. Non-episodic finite-horizon online RL is less standard than the usual episodic or discounted setups, and the paper asks a well-motivated question within this setting: whether one should prefer a target that is easier to learn when interaction is limited.

2. A key conceptual contribution is the surrogate-target perspective itself. The paper is clear that the goal is not to approximate the full-horizon optimal policy as aggressively as possible, but to optimize the trade-off between target quality and learnability. This framing makes the contribution easier to understand than if it were presented as only a technical variant of K-step planning.

3. The theory tells a reasonably coherent story. The paper distinguishes the behavior of fixed-K lookahead from the thresholded surrogate and clarifies why these should be analyzed differently. In particular, it explains why exact short-horizon greedification can remain structurally suboptimal, while thresholding may still be preferable under finite budget because it induces an easier learning problem.

4. The empirical section compares against a reasonably relevant set of tabular and optimistic RL baselines for this problem class, which is more appropriate than generic deep RL baselines would be. This gives the experiments a sensible literature context.

Weaknesses:

1. The main empirical claim is inherently regime-dependent, but the paper does not yet characterize that regime sharply enough. The central message is that a faster-to-learn surrogate can yield higher cumulative reward under limited samples, but the paper gives limited guidance on when this trade-off should be expected to favor thresholding. A stronger characterization of the favorable regime would improve both the conceptual contribution and the practical usefulness of the results.

2. The empirical scope remains somewhat narrow with respect to one of the paper’s own likely dependencies: state revisitation. The proposed advantage appears most plausible when the learner can repeatedly revisit and refine values around the same states, but it is less clear how well the approach transfers to environments where revisitation is weaker or useful information is more diffusely distributed across the state space. Making this dependence more explicit would help readers understand the boundaries of the method.

3. There is a noticeable theory-practice mismatch in the thresholding mechanism. The theoretical analysis is built around LCB-style thresholding, which supports elimination-style arguments and regret control, whereas the experiments use a UCB-based heuristic for action selection. This does not invalidate the empirical results, but it does weaken the extent to which the theory directly explains the practical implementation being evaluated.

---

> ### Author Rebuttal · Authors · 2026-03-30
>
> We thank the reviewer for the thoughtful feedback, and for recognizing the novelty of the surrogate-target perspective, the clarity of our framework, and the relevance of our analysis. We address the concerns below.
>
> **Favorable Regime** We thank the reviewer for raising this point. We will add discussion on the favorable regime to the appendix:
>
> Since the regret lower bound for learning the optimal policy in non-episodic finite horizon is linear, while the regret of LGKT compared with the K-step thresholding policy is sublinear, characterizing the favorable regime reduces to comparing whether the suboptimality gap is smaller than this lower bound.
>
> Further, we will add an instance-dependent suboptimality gap to the revised paper. Please refer to our response to reviewer D7tf for details. As discussed there, empirically, we observe that across all environments considered, faster convergence translates into higher cumulative reward.
>
> **State Revisitation** We thank the reviewer for raising this point. We agree that state revisitation is relevant in tabular online RL. We clarify that this is not a special dependency of our method, but rather a common feature of tabular approaches. For tabular methods, model-free and model-based methods only update the estimator (Q function or P) when a state is revisited. In this sense, revisitation is a fundamental property of the tabular setting, not a limitation introduced by our approach.
>
> Importantly, our theoretical guarantees on convergence to the K-step thresholding policy do not require any revisitation assumption. The key reason is that if a state is visited only a small number of times, then the total regret contributed by that state is limited. Empirically, our environments already include rare revisitation patterns. For example, in FrozenLake, some states are naturally difficult to revisit under certain actions. Despite this, our method still performs strongly.
>
> We will revise the discussion to clarify that revisitation affects all tabular baselines, not uniquely ours, and to better describe how our environments already exhibit weakly revisited states.
>
> **Distinction Between Implemented Algorithm and Theory Version** We appreciate this comment and agree that the distinction should be clarified more carefully. The key point is that the implementation still follows the same thresholding principle as the theory: LCB is used to test whether an action is confidently above the threshold. The only difference arises in the fallback case when no action currently has LCB above the threshold. In the theoretical version, this case uses uniform exploration, while in the implemented version, we instead use UCB-based action selection in this fallback regime.
>
> Thus, the implementation does not replace LCB thresholding with UCB; rather, it preserves LCB for threshold testing and only modifies the fallback exploration rule. We chose this implementation because it is empirically more suitable for high thresholds. When the threshold is set too high, the theoretically faithful version behaves similarly to a random policy. In contrast, the implemented variant still uses optimistic exploration oriented toward reward maximization, which improves performance.
>
> To address this concern directly, we have added additional results comparing the theoretically faithful LCB-style version and the implemented version: [Figure](https://imgur.com/a/ixtl91C). These results show that LG1T and LG2T with the theoretical fallback and a suitably low threshold perform comparably to the implemented algorithm, but are more sensitive to the threshold choice. Specifically, in synthetic instances, using the same thresholds (0.3 for LG1T and 0.9 for LG2T), the theory version is slightly worse than the implemented version but still outperforms benchmarks. In JumpRiverswim with 8 states, the theory version with a smaller threshold (-0.1) performs similarly to the implemented version. Similar behavior holds across other instances and will be included in the revised version. This supports our intended claim: the empirical gains are driven by the thresholding principle itself, while the UCB fallback mainly improves robustness and performance when the threshold is high. We will add this discussion to the Modification paragraph (Line 315) in the revised version.
>
> Additionally, we will include appendix results showing that the convergence rate of the implemented LG1T has the same regret order up to a logarithmic factor. The intuition is in the theoretical version with uniform exploration, we upper bound how often the LCB of a good action falls below the threshold. In the implemented version, though exploration is guided by UCB, we similarly upper bound how often the UCB of a good action is smaller than that of suboptimal actions. This ensures good actions are not persistently dominated by bad ones. As a result, the number of incorrect selections remains controlled up to log factors, leading to the same order of regret.

---

> > ### Author Rebuttal · Reviewer_KRaG · 2026-04-03
> >
> > The rebuttal addresses most of my concerns. In particular, the clarification of the LCB-thresholding vs. UCB-fallback implementation substantially reduces my concern about the theory–practice mismatch, and the discussion of revisitation is also more convincing. I still think the characterization of the favorable regime remains somewhat incomplete, as the rebuttal mainly provides a promising theoretical explanation rather than a fully developed empirical or instance-dependent characterization. Overall, my concerns are mostly, though not fully, resolved.

---

> > > ### Author Response · Authors · 2026-04-03
> > >
> > > We thank the reviewer for noting that our rebuttal helps clarify the empirical advantage of our proposed algorithm. We further elaborate on the notion of the favorable regime.
> > >
> > > To precisely characterize the favorable regime, one needs to compare two quantities: (i) the instance-dependent suboptimality gap of the $K$-step thresholding policy, and (ii) the instance-dependent regret lower bound for learning the optimal policy in finite-horizon non-episodic MDPs. This comparison is natural because we show that LGKT converges to the $K$-step thresholding policy at a sublinear rate. Therefore, our approach is favorable when the suboptimality gap is smaller than the regret required to learn the optimal policy.
> > >
> > > While both quantities are linear in the worst case as stated in Section 2.1 and 3.1, the key difference lies in their instance-dependent constants. That is, both can be written as $c(\text{instance}) \cdot T$, where the constant depends on the specific problem structure. Thus, characterizing the favorable regime requires deriving tight instance-dependent constants for both the suboptimality gap and the regret lower bound, and then comparing them. In other words, one must understand how large the linear coefficient is for each instance.
> > >
> > > This is technically challenging in our setting. In discounted MDPs, such analyses often rely on the contraction property of the Bellman operator. In contrast, in the undiscounted finite-horizon setting we consider, this tool is not available. As a result, one must directly analyze how differences in transition dynamics propagate through the value function, which is substantially more involved.
> > >
> > > Moreover, even deriving tight instance-dependent bounds for regret alone is typically the focus of dedicated work (e.g., [1](https://proceedings.neurips.cc/paper_files/paper/2021/file/000c076c390a4c357313fca29e390ece-Paper.pdf), [2](https://arxiv.org/pdf/2010.03104)). Therefore, fully characterizing the favorable regime, which requires resolving both the suboptimality gap and regret constants simultaneously, is beyond the scope of a single paper.
> > >
> > > Empirically, we will include the following figure ([Figure](https://imgur.com/a/rxoDEmk)) illustrating how the suboptimality gap of the one-step greedy policy varies with the maximum $L_1$ distance for 10 states and 5 actions, since our theoretical result explicitly relates the L1 distance. The horizon length is chosen to be 20000, which aligns with the empirical settings in Section 5. The figure is plotting 300 synthetic instances. We observe a clear trend that aligns with our analysis: as the $L_1$ distance increases, the suboptimality ratio decreases, meaning it is more suboptimal. This provides empirical support for the accuracy of our new suboptimality gap characterization. Moreover, this characterization enables a practical way to identify favorable instances: by comparing the estimated suboptimality gap with the instance-dependent regret lower bound required to learn the optimal policy, one can determine whether our approach operates in a favorable regime. Finally, on the same set of generated instances as shown in Figure 1, we observe that our methods consistently outperform all benchmarks on average, further validating that the favorable regime identified by our analysis translates into improved empirical performance.

---

### Official Review · Reviewer_HLdb · 2026-03-11

**Soundness:** 3
**Presentation:** 3
**Significance:** 3
**Originality:** 3
**Overall Recommendation:** 5
**Confidence:** 3

**Summary:**

This paper studies online reinforcement learning (RL) in non-episodic, finite-horizon settings, where the goal is to maximize cumulative reward up to a known terminal time without the benefit of environment resets. This settings fundamentally differ from conventional infinite-horizon RL and episodic finite-horizon RL settings, and make most of the existing techniques not directly applicable or sample in-efficient. To addresses these issues, this paper first introduces a K-step lookahead thresholding policy. Leveraging this threshold policy, the authors further develop LGKT (LCB-Guided K-Step Thresholding), which is a new online algorithm for non-episodic, finite horizon RL. Theoretical performance analysis in terms of regret is presented for LGKT, along with empirical validations on a suite of non-episodic environments.

**Compliance With Llm Reviewing Policy:**

Affirmed.

**Final Justification:**

This paper is theoretically sound and makes an important contribution in this domain. The authors' rebuttals further help me understand this paper and I support the acceptance of this work.

**Key Questions For Authors:**

- In Section 4.1, the authors mentioned that the choice of $g(t)$ ensures that the LCB of truly sub-threshold actions drops below $\gamma_t$ rapidly. Can you elaborate a bit on this?
- this paper discussed the choice of the hyperparamter $\gamma$ in Appendix C.2. and demonstrate its robustness in the experiments. Will this parameter be adaptively tuned to achieve the "fundamental trade-off"?

**Limitations:**

Yes.

**Strengths And Weaknesses:**

Strength:
- This paper studied an important and practical setting, i.e., non-episodic, finite-horizon MDP, which has many real-world applications, but less studied compared to the popular infinite-horizon RL and episodic finite-horizon RL settings.
- This new settings has unique property or characters that make many conventional techniques for infinite-horizon RL and episodic finite-horizon RL settings either not directly applicable or become sample in-efficient.
- This paper develops a K-step lookahead thresholding policy, which is a new class of designed for sample-efficient learning in non-episodic finite-horizon MDPs. This threshold policy is provably optimal when $K$ exceeds the total horizon $T$.
- One of the key algorithm is the LGKT (LCB-Guided K-Step Thresholding), for the non-episodic, finite horizon RL. LGKT leverages the threshold policy for online decision makings. More importantly, LGKT is provably guaranteed with a minimax optimal regret when $K=1$ and a sublinear regret when $K\geq 2$. This significantly improves the state-of-the-art methods in this new setting, which only offers a linear regret.
- The performance of LGKT is evaluated in three interesting practical environments.

Weakness:
- In Section 4.1, the authors mentioned that the choice of $g(t)$ ensures that the LCB of truly sub-threshold actions drops below $\gamma_t$ rapidly. Can you elaborate a bit on this?
- this paper discussed the choice of the hyperparamter $\gamma$ in Appendix C.2. and demonstrate its robustness in the experiments. Will this parameter be adaptively tuned to achieve the "fundamental trade-off"?

---

> ### Author Rebuttal · Authors · 2026-03-30
>
> We thank the reviewer for the positive assessment, particularly for recognizing the importance of the non-episodic finite-horizon setting, the novelty of the $K$-step lookahead thresholding policy, the strong regret guarantees of LGKT, and the practical relevance of our experiments. We address the questions below.
>
> **The Role of $g(t)$** We thank the reviewer for raising this point. We will add further discussion in the revision to clarify the role of the lower confidence bound and the bonus term.
>
> Intuitively, using an LCB tends to eliminate actions that are either insufficiently explored or whose reward is genuinely below the threshold. The reason is that, after subtracting the confidence radius, an action can only remain above the threshold if its true reward is sufficiently separated from the threshold to compensate for this downward shift. Hence, actions whose reward is below the threshold, or not sufficiently above it, will tend to have LCB below $\gamma$ and be filtered out quickly.
>
> Moreover, our choice of bonus term $g(t)$ depends on the visit count of the state--action pair rather than the global time index. This is important because it allows the confidence radius to shrink according to how much information has actually been collected about that state--action pair. In particular, for good actions, it does not matter if it is only visited enough times when the horizon is large. Since the visit count is typically much smaller than the current time, once a promising action has been visited enough times, its confidence radius becomes small, and its LCB can rise above the threshold. This leads to a more data-efficient distinction between actions. We will revise Section 4.1 to make this intuition more explicit.
>
> **Choice of Threshold** We agree that adaptively tuning the threshold parameter is an important direction, and potentially a way to further unleash the benefit of thresholding. One natural idea is to start from a relatively low threshold and then gradually increase it as the learner gains enough information about the rewards of different actions. In this way, we can ensure a fast convergence in the beginning and warm start learning a better policy. We conjecture that a valid data-driven rule could be for each state, if the number of time that state is visited exceed $\log(T)$ after the last change of threshold, increase the threshold by 0.3. As seen in the figure ([Figure](https://imgur.com/a/wnlnwvx)), for synthetic and frozenlake environments, this heuristic (LG1T-I) allows us to start LG1T from initial threshold being 0 for all states while the overall performance could exceed the performance of starting at higher threshold. We omitted results for other instances because Figures 7-8 already showed that algorithms perform the same for all thresholds.
>
> At the same time, our current theoretical results, including Theorems 4.2 and 4.4, do not require the threshold to be the same across the entire horizon and for each state. Thus, in principle, the framework is compatible with horizon and state-dependent threshold choices. In this paper, however, our primary goal is to first identify a quickly converging online algorithm for the K-step thresholding policy and to analyze it rigorously. Empirically, we already observe that even with a simple non-adaptive threshold choice, our method outperforms all benchmarks. We therefore view a rigorous way of data-driven threshold selection as an important and promising future direction.

---

> > ### Author Rebuttal · Reviewer_HLdb · 2026-04-02
> >
> > Thank you for the detailed explanations. This helps me better understand this paper. I will keep the current positive score.

---

> > > ### Author Response · Authors · 2026-04-02
> > >
> > > Thank you for your thoughtful and positive comments!

---

### Official Review · Reviewer_Y1Vp · 2026-03-11

**Soundness:** 3
**Presentation:** 3
**Significance:** 3
**Originality:** 3
**Overall Recommendation:** 5
**Confidence:** 3

**Summary:**

The paper addresses the online learning problem in finite horizon MDPs, where the agent experiences a single, non-repeating trajectory of interactions and needs to maximize rewards over that trajectory.
The authors introduce a k-step lookahead Q function suitable for online non-episodic finite horizon MDPs. They then introduce a thresholding mechanism for action selection, and develop a tabular algorithm for learning the k-step lookahead function. They prove that for the case of k=1, the algorithm achieves minimax optimal constant regret. Finally, they evaluated the algorithm across a few tabular MDPs showing promising results.

**Compliance With Llm Reviewing Policy:**

Affirmed.

**Final Justification:**

I maintain my positive support for this paper as the rebuttal had addressed the concerns I originally had.

**Key Questions For Authors:**

- How sensitive is the LG1-2T switching time t_c to the environment?
- Could the thresholding idea be combined with model-based methods to get both fast initial convergence and asymptotic optimality without a fixed switching time?
- In the K-step case, the exploration probability ε_t triggers the ALG_{K-1} subroutine O(K√T) times total. What is the wall-clock overhead of running the subroutine versus the main thresholding loop?

**Limitations:**

no. I think the main limitation is around the sub-optimality gap.

**Strengths And Weaknesses:**

## Strengths:
- The online learning problem in finite-horizon MDP is well motivated and is underexplored.
- The core idea of learning a k-step lookahead Q function is intuitive and elegant.
- The paper has strong theoretical results, mainly theorem 4.2 on the lower bound for the case of K=1 and the general results for any k (theorem 4.4). The negative results in theorem 3.4 on the optimality gap is also important.
- The tabular experiments are thorough and include many baselines.



## Weaknesses
- Theorem 3.3 only holds for a two-state MDP and under the assumption of stochastic dominance which is restrictive. It's unclear how often real-world problems satisfy this condition, and the paper doesn't provide discussion on what happens in general multi-state MDPs without this assumption.
- It doesn’t seem that there is a principled way of choosing \gamma. The ablations in Appendix E.3 are useful here but an automated way of finding \gamma would be beneficial.
- The regret in equation 4 compares against non-optimal policy rather than the optimal policy.
- The optimality gap in Theorem 3.4 is quite significant.

---

> ### Author Rebuttal · Authors · 2026-03-30
>
> We thank the reviewer for the positive evaluation, particularly for recognizing the importance of the non-episodic finite-horizon setting, the intuitive design of the K-step lookahead formulation, the strength of our theoretical results, and the thorough experimental validation. We address the reviewer’s concerns and questions below.
>
> **Suboptimality Gap** We thank the reviewer for raising this point. Our intent in Theorem 3.3 is not to claim that this assumption holds broadly, but to provide one concrete setting where the thresholding policy can be shown to be optimal. For the more general case, Theorem 3.4 shows that the worst-case suboptimality gap can be linear. To illustrate the practicality of Assumption 3.2 (stochastic dominance), we will add the following example in the revised version: in healthcare decision-making, higher treatment intensity (action) can both increase immediate reward (e.g., higher recovery probability) and shift the next-state distribution toward healthier states.
>
> Additionally, we will add an instance dependent suboptimlaity gap to the revised paper. Please refer to our response to reviewer D7tf for details. As discussed there, more tightly characterizing the gap is difficult and beyond the scope of this paper, as a significant portion of this paper is devoted to the learning part, and establishing our algorithm design translates to a higher cumulative reward empirically.
>
> **Choices of Threshold**We have designed a heuristic method for adaptively tuning the threshold; please refer to our response to reviewer HLdb for details. We agree that adaptively tuning the threshold is an important direction to further improve LGKT, and we leave a more systematic evaluation to future work.
>
> **Regret** Our regret is defined with respect to the $K$-step thresholding policy rather than the globally optimal policy to illustrate that our algorithm admits faster non-episodic finite-horizon convergence. As highlighted in Section 2.1, algorithms aiming to achieve optimality in finite-horizon non-episodic settings face a linear lower bound, so proving a regret bound against the global optimum would not meaningfully capture the advantage of faster convergence to a strong surrogate. The main contribution is to show that in this challenging setting, converging quickly to a surrogate policy can yield higher cumulative reward than slowly converging to the exact optimum—a point we validate empirically in Section 5. We will revise the paper to clarify that our regret definition is intentionally designed to reflect this faster finite-sample performance.
>
> **Switching Time** The performance is sensitive with switching time because the benefit of switching to LG2T will only be evident when one step reward is learned accurately. As shown in this additional figure ([Figure](https://imgur.com/a/dURoUBe)) on experiments under synthetic and an 8-state JumpRiverswim environment, switching too early when 1-step reward is not learned accurately, the algorithm will be no worse than LG2T but worse than LG1T. However, when switching after 1-step reward is learned accurately, the algorithm will outperform both LG1T and LG2T. In the current paper, we use a fixed switching rule for simplicity. We view designing an adaptive switching criterion as an important future direction. One possible data-driven rule would be to initialize the LG2T procedure after $\sqrt{SAT}$. This is because this matches the worst case regret upper bound of LG1T, which means after this, LG1T should learn 1-tep reward accurately. We show in an additional experiment ([figure](https://imgur.com/a/QMZwmWr)) that this heuristic (LG1-2T) could outperform both LG1T and LG2T in 100 state and JumpRiverswim environments. We will add the results for all environments to the appendix of the revised paper.
>
> **Changing to RL Algorithm** This is possible and an important next step. A natural criterion could be based on whether enough state-action pairs have been visited sufficiently often so that the model-based method can make effective use of the accumulated knowledge after the switch. However, as shown in Figures 1-3, RL algorithms perform badly within 20000 steps. We think testing numerically whether the heuristic is beneficial is out of the scope of this paper, as the benefit would only appear after a sufficiently long horizon, which is not the focus of the current setting. We will revise our conclusion to include this point.
>
> **Wall Clock of Subroutine** This depends on the specific subroutine used. For LG2T, where the subroutine is UCB, each call only requires computing the corresponding UCB. Therefore, the per-iteration computational cost of the subroutine is of the same constant order as that of the main loop. We reported the confidence interval of the number of times over 100 repetitions that the subroutine is triggered for synthetic environments: 10 state:[317, 578]. For other environments, they all scaled similarly as $\Theta(K\sqrt{SAT})$.

---

> > ### Author Rebuttal · Reviewer_Y1Vp · 2026-04-02
> >
> > Thank you for the rebuttal. It clarified my understanding.

---

> > > ### Author Response · Authors · 2026-04-02
> > >
> > > Thank you for your thoughtful and positive comments!

---

### Decision · Program_Chairs · 2026-04-30

**Decision:**

Accept (regular)

**Comment:**

This submission introduces a $K$-step lookahead thresholding mechanism for online reinforcement learning in non-episodic, finite-horizon Markov Decision Processes (MDPs). The authors target the challenge of estimating returns to a fixed terminal time—a structure often neglected by infinite-horizon methods—by proposing a modified Q-function that truncates planning to $K$ steps.

The reviewers initially raised several questions regarding the theoretical framework, specifically seeking a clearer relationship between the proposed greedy and thresholding policies. There were also concerns about the experimental setup and the applicability of finite-sample convergence results to infinite-horizon settings. During the rebuttal phase, the authors provided a clarifying response and  these additions provided the necessary intuition and technical justification to satisfy the reviewers, leading to score increases. The consensus is that the work represents a meaningful extension of RL theory with both practical and theoretical merits. Consequently, the paper is recommended for acceptance.

Recommendation: Accept.